# EdgeCrafter: Compact ViTs for Edge Dense Prediction via Task-Specialized Distillation

## Abstract

Deploying high performance dense prediction models on resource-constrained edge devices remains challenging due to strict limits on computation and memory. In practice, lightweight systems for object detection, instance segmentation, and pose estimation are still dominated by CNN-based architectures such as YOLO, while compact Vision Transformers (ViTs) often struggle to achieve similarly strong accuracy–efficiency trade-offs, even with large scale pretraining. We argue that this gap is largely due to insufficient task-specific representation learning in small-scale ViTs, rather than an inherent mismatch between ViTs and edge dense prediction. To address this issue, we introduce EdgeCrafter, a unified compact ViT framework for edge dense prediction centered on ECDet, a detection model built from a distilled compact backbone and an edge-friendly encoder–decoder design. We first adapt a large DINOv3 pretrained ViT to object detection and use it as a task-specialized teacher to distill rich representations into compact student backbones. We further improve efficiency by replacing standard patch embedding with a lightweight convolutional stem and constructing multi-scale features with simple interpolation and linear projection instead of costly feature pyramids. The resulting detection-distilled representation transfers directly to instance segmentation and human pose estimation through lightweight task-specific prediction modules. On the COCO dataset, ECDet-S achieves 51.7 AP with fewer than 10M parameters using only COCO annotations. For instance segmentation, ECInsSeg achieves performance comparable to RF-DETR-Seg while using substantially fewer parameters and without the need for additional Objects365 pretraining. For pose estimation, ECPose-X reaches 74.8 AP, significantly outperforming YOLO26-Pose-X (71.6 AP). These results show that compact ViTs, when paired with task-specialized distillation and edge-aware design, can be a practical and competitive option for edge dense prediction. The code and pretrained models for reproducing our results will be released upon publication.

## 1 Introduction

Object detection (Girshick, 2015), instance segmentation (He et al., 2017), and human pose estimation (Cao et al., 2019) are core dense prediction tasks in computer vision, with applications ranging from autonomous driving (Arnold et al., 2019) and robotics (Labbé et al., 2020) to augmented reality (Chan et al., 2019) and video analytics (Caetano et al., 2019). Bringing these models to edge devices is attractive for reasons such as responsiveness and privacy, but it remains difficult because edge platforms operate under tight resource budgets. Beyond computation, edge deployment is also constrained by limited memory budgets, which makes parameter-efficient models particularly important in practice. These constraints are especially pronounced in battery-powered or embedded settings, where both model size and compute cost directly affect deployability.

In real-time edge deployment scenarios, dense prediction tasks are still overwhelmingly dominated by models built upon convolutional neural network (CNN) backbones. The YOLO family (Redmon et al., 2016; Redmon & Farhadi, 2017; Glenn., 2023; 2024; Li et al., 2022a; Wang et al., 2024b;a) and DETR-based models (Lv et al., 2024; Peng et al., 2025) are representative examples, offering strong accuracy–efficiency trade-offs. While pioneering attempts have explored adopting Vision Transformers (ViTs) as backbones for dense prediction (Chen et al., 2024; Li et al., 2022c), these approaches typically rely on scaling to large model sizes

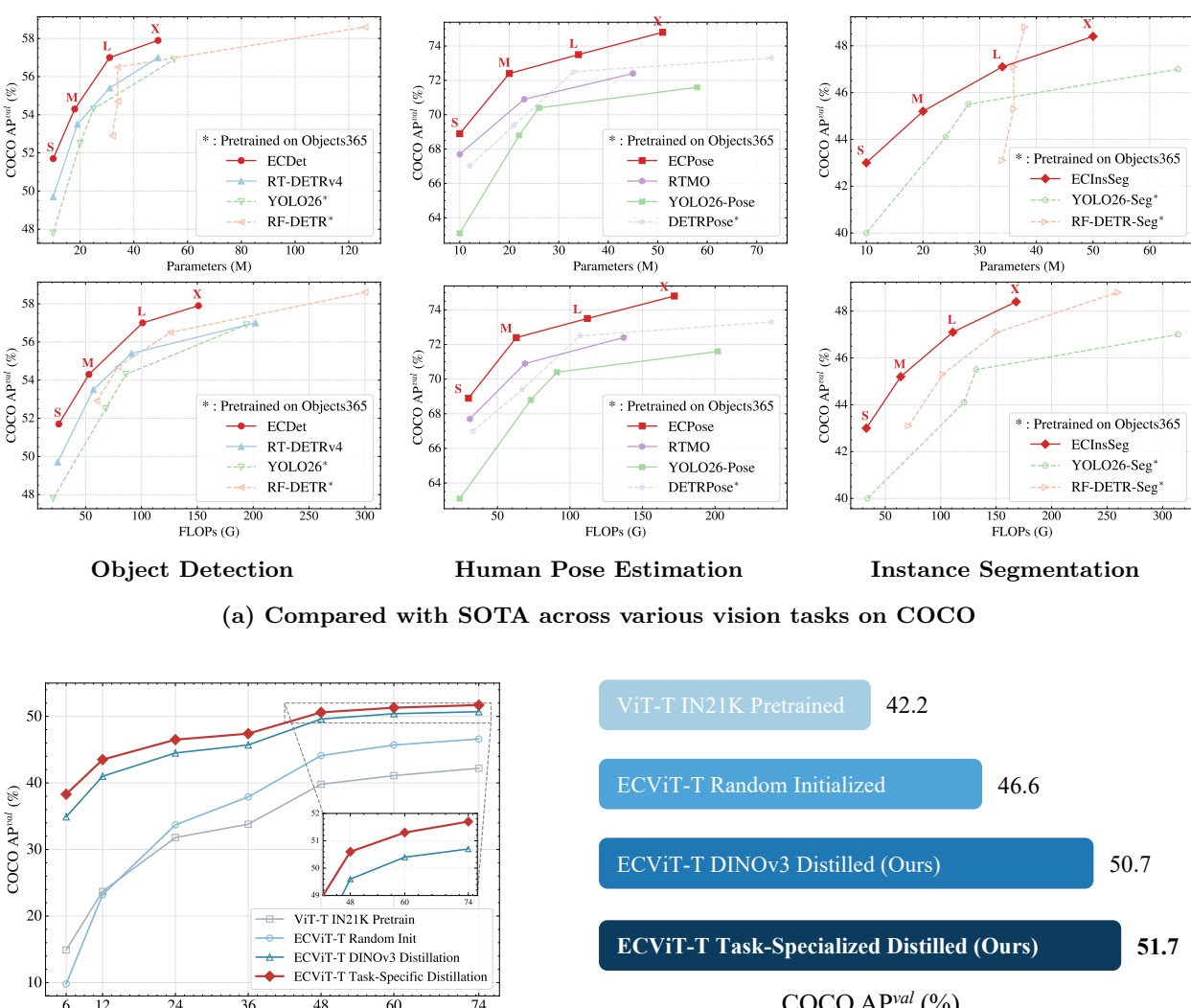

**(a) Compared with SOTA across various vision tasks on COCO**

**(b) Effect of pretraining and distillation strategies for the backbone**

Figure 1: **Comprehensive evaluation of EdgeCrafter.** (a) Comparison with state-of-the-art methods across multiple vision tasks on COCO (Lin et al., 2014). The plots show model parameters (top row) and FLOPs (bottom row) versus mAP. Methods marked with * are pre-trained on the Objects365 dataset (Shao et al., 2019). From left to right, the columns correspond to object detection, human pose estimation, and instance segmentation. (b) Analysis of different pretraining strategies for the backbone, based on the ECDet-T model. ViT-T (Tiny) follows the training strategy proposed in (Steiner et al., 2022) on ImageNet-21K (Ridnik et al., 2021). In our experiments, supervised ImageNet-21K pretraining is weaker than no pretraining for this compact model, consistent with observations reported by Ghiasi et al. (2021); Zoph et al. (2020). Task-specialized distillation yields substantially stronger downstream performance.

(e.g., ViT-Base or Large) or require additional massive pretraining datasets such as Objects365 (Shao et al., 2019) to achieve competitive performance. Recently, large-scale self-supervised pretraining paradigms (Oquab et al., 2023; Siméoni et al., 2025) have unlocked the extraordinary representation capabilities of ViTs. Recent works such as RF-DETR (Robinson et al., 2026) directly employ DINOv2 (Oquab et al., 2023) as a backbone. However, even the smallest variant contains approximately 30M parameters, imposing prohibitive computational and memory constraints for strict edge deployment.

Unfortunately, naively shrinking a ViT to meet stringent edge constraints results in a precipitous drop in performance. Our empirical observations show that even initializing compact ViTs with supervised IN-21K (Ridnik et al., 2021; Steiner et al., 2022) pretraining [1] yields unsatisfactory performance for dense prediction, sometimes even performing worse than training from scratch (Figure 1b). This observation, which is consistent with findings reported by Ghiasi et al. (2021); Zoph et al. (2020), suggests that generic supervised pretraining alone is often insufficient for compact ViTs in edge dense prediction.

In this work, we introduce *EdgeCrafter*, a unified compact ViT framework for edge dense prediction centered on ECDet. Our approach is built upon two key ideas. First, we employ task-specialized knowledge distillation: a large DINOv3-pretrained ViT (Siméoni et al., 2025) is first adapted for object detection and then used as a teacher to supervise compact student models. Second, we design an edge-friendly student architecture that replaces the standard patch embedding with a lightweight convolutional stem (Xiao et al., 2021) and constructs multi-scale features using simple interpolation and linear projections. The resulting distilled backbone and encoder are used directly in ECDet and can be transferred to ECInsSeg and ECPose through lightweight task-specific heads, resulting in a unified framework for object detection, instance segmentation, and human pose estimation.

Figure 1 summarizes the two main messages of this work. First, Figure 1a shows that EdgeCrafter achieves strong accuracy-efficiency trade-offs across object detection, instance segmentation, and human pose estimation, measured by parameters and FLOPs, while reusing a common detection-distilled representation. Several competitive baselines in these comparisons also rely on additional Objects365 pretraining, yet EdgeCrafter remains competitive and in some cases outperforms them using only task-specific COCO annotations. Second, Figure 1b shows that the representation learning strategy matters substantially: standard generic pretraining is not enough for compact ViTs, whereas task-specialized distillation leads to much stronger downstream performance. Together, these results support the view that compact ViTs can be competitive for edge dense prediction when their representations and architecture are designed for the target setting.

In summary, this work makes the following contributions:

- We identify insufficient task-specific representation learning as a key bottleneck for compact ViTs in edge dense prediction, and show that generic supervised pretraining alone is often inadequate for small models.

- We propose an edge-oriented compact ViT design that combines task-specialized distillation, a lightweight convolutional stem, and simple multi-scale feature construction, making ViT backbones more suitable for dense prediction under tight parameter and FLOP budgets.

- We introduce EdgeCrafter, a unified framework centered on ECDet, and show that its detection-distilled representation transfers effectively to instance segmentation and human pose estimation while maintaining strong cross-task accuracy-efficiency trade-offs.

Our code will be released upon publication.

## 2 Related Work

**Knowledge distillation from vision foundation models.** Knowledge distillation (KD) (Hinton et al., 2015; Kim et al., 2018; Ba & Caruana, 2014; Mirzadeh et al., 2020; Beyer et al., 2022) is a standard tool for transferring knowledge from large models to smaller ones, either through logits or intermediate features (Romero et al., 2015; Huang & Wang, 2017; Ahn et al., 2019; Heo et al., 2019; Zagoruyko & Komodakis, 2017; Sun et al., 2021; Wei et al., 2022). With the emergence of vision foundation models, recent work has explored distillation from CLIP (Radford et al., 2021), SAM (Kirillov et al., 2023), DINOv2 (Oquab et al., 2023), and DINOv3 (Siméoni et al., 2025) into smaller backbones or unified representations (Wei et al., 2022; Wang et al., 2024c; Ranzinger et al., 2024). These studies show that powerful pretrained teachers can improve downstream transfer, but they are mostly designed for general-purpose representation learning rather

---

[1] https://huggingface.co/timm/vit_tiny_patch16_224.augreg_in21k

than compact edge-oriented dense prediction. In contrast, our goal is to convert a large pretrained ViT into a task-specialized teacher for dense prediction, and then distill that knowledge into compact students that remain effective under tight parameter and FLOP budgets.

**Efficient object detection.** Efficient object detectors are still dominated in practice by CNN-based families such as YOLO (Redmon et al., 2016; Redmon & Farhadi, 2017; 2018; Li et al., 2022a; Wang et al., 2024b;a; Glenn., 2023; 2024; Tian et al., 2025), which provide strong accuracy-efficiency trade-offs and broad task coverage. DETR (Carion et al., 2020) introduced end-to-end set prediction for detection, and subsequent work improved convergence and efficiency (Zhu et al., 2021; Liu et al., 2022; Li et al., 2022b; Zhang et al., 2023; Zhao et al., 2024; Lv et al., 2024; Peng et al., 2025; Wang et al., 2025). More recent detectors have combined this line of work with ViT backbones and large-scale pretraining. LW-DETR (Chen et al., 2024) uses a pretrained ViT backbone, RF-DETR (Robinson et al., 2026) further incorporates DINOv2 (Oquab et al., 2023) and architecture search, DEIMv2 (Huang et al., 2025a) combines DINOv3 (Siméoni et al., 2025) with lightweight convolutions, and RT-DETRv4 (Liao et al., 2025) aligns detector training with DINOv3 representations. These works demonstrate the value of pretrained transformer features, but strong performance often depends on comparatively large backbones or additional large-scale pretraining. Our work instead focuses on transferring task-specialized ViT representations into compact students for edge dense prediction.

**Efficient instance segmentation.** Instance segmentation methods have evolved from two-stage frameworks such as Mask R-CNN (He et al., 2017) to more efficient one-stage or end-to-end designs, including YOLACT (Bolya et al., 2019), BlendMask (Chen et al., 2020), SOLO (Wang et al., 2020), and FastInst (Wu et al., 2023). In parallel, query-based transformer methods such as MaskFormer (Cheng et al., 2021), Mask2Former (Cheng et al., 2022), and MaskDINO (Li et al., 2023) have shown that end-to-end segmentation can benefit from strong dense representations. RF-DETR-Seg (Robinson et al., 2026) further extends the pretrained RF-DETR framework to efficient instance segmentation. Relative to this line of work, our goal is not to introduce a new segmentation-specific backbone, but to show that a compact distilled ViT backbone can support instance segmentation effectively within the same unified framework used for detection and pose estimation.

**Efficient human pose estimation.** Human pose estimation methods include top-down and bottom-up pipelines, as well as more recent end-to-end set prediction approaches. Practical efficient systems often build on CNN-based or YOLO-style detectors (Lu et al., 2024; Glenn., 2024; Maji et al., 2022), while DETR-based pose estimators such as PETR (Shi et al., 2022), QueryPose (Xiao et al., 2022), ED-Pose (Yang et al., 2023), GroupPose (Liu et al., 2023), and DETRPose (Janampa & Pattichis, 2025) adapt set-based prediction to multi-person keypoint estimation. These methods show that transformer decoders can be effective for pose estimation, especially when paired with strong detection backbones. Our work is complementary: we focus on the backbone side of the problem and show that a compact distilled ViT can serve as a shared representation for pose estimation, while also supporting object detection and instance segmentation in the same framework.

## 3 Method

Our approach is illustrated in Figure 2. We first adapt a pretrained DINOv3 model (Siméoni et al., 2025) to object detection and use it as a task-specialized teacher. We then distill its representations into a compact ViT backbone, denoted ECViT, and build ECDet on top of the distilled backbone. Finally, we reuse the same distilled backbone and encoder in the instance segmentation model ECInsSeg and the human pose estimation model ECPose by attaching lightweight task-specific heads. This yields a unified framework in which detection is the primary training setting, while the learned representation transfers to other dense prediction tasks.

### 3.1 Overview

ECDet is the main instantiation of EdgeCrafter. It combines a compact ViT backbone with an RT-DETR-style encoder-decoder detector (Lv et al., 2024). The backbone is adapted to dense prediction in two ways: we

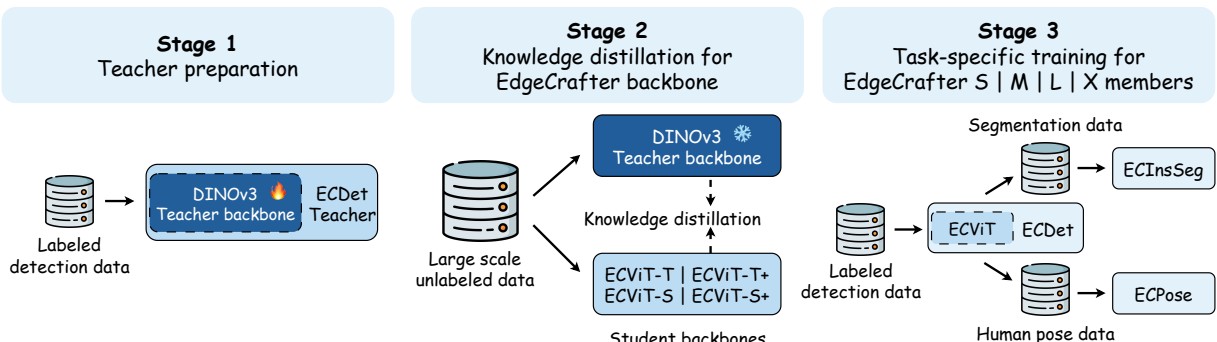

Figure 2: **Overview of the EdgeCrafter pipeline. Stage 1:** A pretrained DINOv3 backbone (Siméoni et al., 2025) is adapted to object detection to create a task-specialized teacher within the ECDet formulation. **Stage 2:** The resulting teacher distills its detection-oriented representation into compact ECViT student backbones through feature alignment on a large image collection. **Stage 3:** The distilled students are used to instantiate the ECDet model family at different scales (S/M/L/X), and the same distilled backbone and encoder are further reused for instance segmentation and human pose estimation with lightweight task-specific heads. The key idea is that detection serves as the representation-learning stage, while the learned backbone transfers directly to other dense prediction tasks.

replace the standard large-stride patch embedding with a convolutional stem that preserves local structure, and we construct a lightweight multi-scale feature pyramid from the final transformer blocks instead of using a heavy pyramid module. To make this compact backbone effective, we distill it from a detection-specialized teacher using feature alignment. The resulting representation is then reused by ECInsSeg and ECPose, which share the same backbone and encoder but use task-specific prediction modules.

## 3.2 ECDet Architecture

**ECDet** consists of a compact ViT backbone, denoted ECViT, followed by an encoder and a decoder for set-based object prediction. Figure 3 summarizes the architecture.

**Backbone design.** Compact ViTs for detection must preserve local structure while remaining efficient. A standard ViT patch embedding applies a single large-stride projection at the input, which is effective for image classification but can discard fine spatial details that matter for dense localization. Following prior work that replaces patch embedding with a convolutional stem (Xiao et al., 2021), we use a stack of four $3 \times 3$ convolutions with stride 2 in the ECDet backbone. Our motivation is detection-specific: instead of aggregating local information in one step, the stem enlarges the receptive field progressively before the transformer blocks. This is also consistent with effective receptive field analyses (Luo et al., 2016), which show that stacked convolutions retain a center-concentrated effective receptive field even when their theoretical receptive field grows. We therefore use a moderate receptive field in the stem and analyze this choice in the detection ablations.

**Multi-scale feature generation.** ViTs do not naturally produce hierarchical feature pyramids, so we construct one explicitly from the final transformer blocks. Let $\mathbf{X}_{L-1}, \mathbf{X}_L \in \mathbb{R}^{(\frac{H}{16} \frac{W}{16}) \times C}$ denote the output tokens from the last two transformer blocks. We first average and reshape them into a spatial feature map at stride 16:

$$\mathcal{F}^{(16)} = \frac{1}{2} \left( \mathbf{X}_{L-1} + \mathbf{X}_L \right),$$

and then form a three-level pyramid $\{\mathcal{F}^{(8)}, \mathcal{F}^{(16)}, \mathcal{F}^{(32)}\}$ with bilinear interpolation $\mathcal{B}_s(\cdot)$ and a $1 \times 1$ convolutional projection $\Theta_s(\cdot)$:

$$\mathcal{F}^{(s)} = \Theta_s\big(\mathcal{B}_s(\mathcal{F}^{(16)})\big), \quad s \in \{8, 16, 32\}.$$

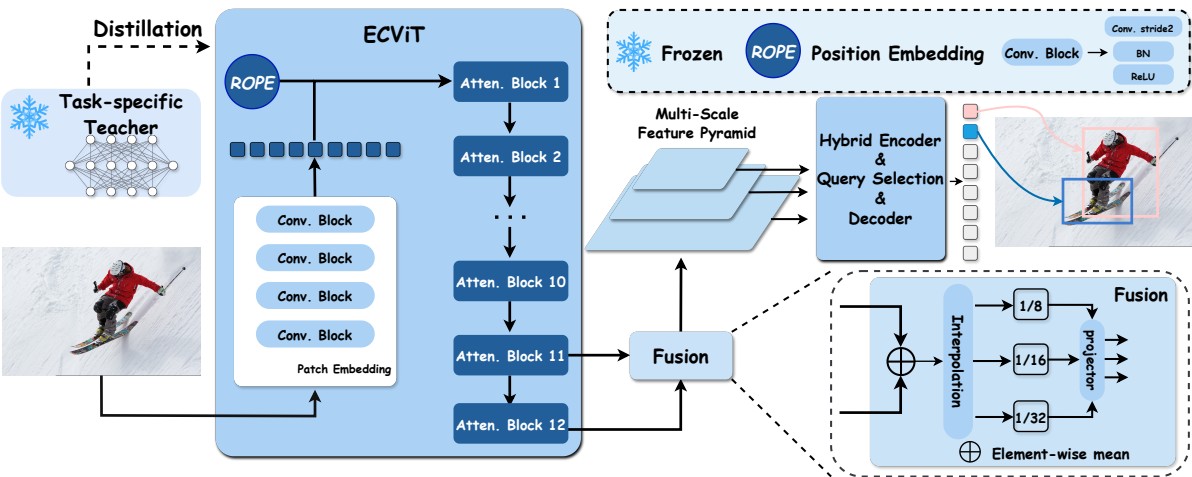

Figure 3: **Architecture of ECDet.** ECDet consists of three components: a distilled ECViT backbone, an encoder, and a decoder. The backbone replaces the standard large-stride patch embedding with a four-layer convolutional stem and outputs a single-resolution token representation. A lightweight multi-scale feature generator then aggregates the final transformer blocks and produces feature maps at strides 8, 16, and 32 with interpolation and $1 \times 1$ projections. The encoder refines and fuses these features, and the decoder performs set-based object prediction from learned object queries. The overall design keeps the detector compact while preserving the multi-scale structure required for dense localization.

This design is deliberately simple and introduces very few additional parameters beyond the backbone itself, which is desirable for edge dense prediction. At the same time, it supplies the detector with the multi-scale features needed for dense localization without resorting to heavier feature pyramid modules.

**Encoder.** Our encoder follows RT-DETR (Lv et al., 2024). The coarsest feature map is first refined by attention-based intra-scale feature interaction (AIFI) (Lv et al., 2024), which applies self-attention on the lowest-resolution feature map to enlarge its receptive field and strengthen long-range contextual interactions at modest cost:

$$\hat{\mathcal{F}}^{(32)} = \mathrm{AIFI}\big(\mathcal{F}^{(32)}\big).$$

The refined coarse feature is then fused with the higher-resolution features through CNN-based cross-scale feature fusion (CCFF) (Lv et al., 2024), which propagates the high-level context from $\hat{\mathcal{F}}^{(32)}$ back to the finer scales and combines them with local spatial detail:

$$\mathcal{F}_{\mathrm{enc}} = \mathrm{CCFF}\big(\hat{\mathcal{F}}^{(32)}, \mathcal{F}^{(16)}, \mathcal{F}^{(8)}\big).$$

The encoded representation $\mathcal{F}_{\mathrm{enc}}$ provides the shared dense representation used by ECDet and by the lightweight extensions to segmentation and pose.

**Decoder.** The decoder follows the DETR set-prediction formulation (Carion et al., 2020), with stacked self-attention, deformable cross-attention (Zhu et al., 2021), and feed-forward layers operating on learned object queries $\mathbf{Q} \in \mathbb{R}^{N \times C}$. We use four decoder layers for all ECDet variants and fix the number of queries to $N = 300$, which keeps the detector compact while preserving strong set-based prediction performance.

**Training objective.** ECDet is trained with a standard DETR-style bipartite matching objective. Specifically, the total detection loss is defined as:

$$\mathcal{L}_{\mathrm{det}} = \lambda_{\mathrm{cls}}\mathcal{L}_{\mathrm{cls}} + \underbrace{\lambda_{\ell_1}\mathcal{L}_{\ell_1} + \lambda_{\mathrm{giou}}\mathcal{L}_{\mathrm{giou}} + \lambda_{\mathrm{ddf}}\mathcal{L}_{\mathrm{ddf}} + \lambda_{\mathrm{fgl}}\mathcal{L}_{\mathrm{fgl}}}_{\text{box regression losses}},$$

Table 1: **ECDet model configurations.** The four detector variants share the same overall architecture and input resolution ($640 \times 640$), while scaling model capacity through the ECViT backbone width, encoder hidden dimensions, and decoder FFN dimensions. 'Variant' denotes the ECViT backbone type used in each model. 'Teacher' indicates the teacher backbone used during distillation. All models use a four-layer decoder and 300 object queries.

| Model | Resolution | ECViT Backbone | | | | | Encoder | | Decoder | | |
| | | Variant | Embed Dim | Attn Heads | FFN Ratio | Teacher | Hidden Dim | FFN Dim | Layers | FFN Dim | Queries |
|---|---|---|---|---|---|---|---|---|---|---|---|
| S | 640 | T | 192 | 3 | 4 | ECTeacher-S | 192 | 512 | 4 | 512 | 300 |
| M | 640 | T+ | 256 | 4 | 4 | ECTeacher-B | 256 | 512 | 4 | 1024 | 300 |
| L | 640 | S | 384 | 6 | 4 | ECTeacher-B | 256 | 1024 | 4 | 1024 | 300 |
| X | 640 | S+ | 384 | 6 | 6 | ECTeacher-B | 256 | 2048 | 4 | 2048 | 300 |

where $\mathcal{L}_{\mathrm{cls}}$ denotes the classification loss (Huang et al., 2025b). The terms $\mathcal{L}_{\ell_1}$ and $\mathcal{L}_{\mathrm{giou}}$ represent the standard $\ell_1$ distance and Generalized IoU losses (Rezatofighi et al., 2019) for bounding box coordinate regression. To further enhance localization precision, we incorporate the Decoupled Distillation Focal (DDF) loss $\mathcal{L}_{\mathrm{ddf}}$ and Fine-Grained Localization (FGL) loss $\mathcal{L}_{\mathrm{fgl}}$ following the D-FINE paradigm (Peng et al., 2025), which refine the box boundary distributions. The coefficients $\lambda_{\mathrm{cls}}$, $\lambda_{\ell_1}$, $\lambda_{\mathrm{giou}}$, $\lambda_{\mathrm{ddf}}$, and $\lambda_{\mathrm{fgl}}$ are hyper-parameters used to balance the contribution of each task.

**Model scaling and architecture details.** We instantiate ECDet at four scales, denoted **S**, **M**, **L**, and **X**, all evaluated at an input resolution of $640 \times 640$. Table 1 summarizes the configurations. Across this family, we keep the overall detector topology fixed and scale the model primarily through backbone and hidden dimensions. Specifically, the ECViT backbone uses four variants, T, T+, S, and S+, whose embedding dimensions increase from 192 to 384, whose number of attention heads increase from 3 to 6, and whose FFN expansion ratios increase from 4 to 6. The detector matches the encoder and decoder widths to the backbone capacity. The encoder hidden dimension grows from 192 to 256, while the encoder FFN dimension increases from 512 to 2048. The decoder always uses four layers and 300 learned object queries, but its FFN dimension is scaled from 512 to 2048 across the model family. We pair each student scale with a compatible teacher during distillation: the smaller S variant uses ECTeacher-S, while the larger M, L, and X variants use ECTeacher-B.

### 3.3 Task-Specialized Distillation

**Detection-specialized teacher.** We build the teacher by adapting a pretrained DINOv3 model (Siméoni et al., 2025) to object detection with the same detector formulation used by ECDet. This converts a general-purpose foundation model into a detection-specialized teacher whose representation is directly aligned with the downstream student task. In the final model family, we use two teacher scales: ECTeacher-S and ECTeacher-B, obtained by adapting DINOv3-S and DINOv3-B, respectively. The details of ECTeacher are provided in Section A.1. These teacher scales provide a practical balance between teacher strength and student-teacher compatibility, and the influence of teacher scale is analyzed in the ablations.

**Feature-alignment distillation.** The student backbone is trained to match the teacher representation through one-to-many feature alignment. Let $\mathbf{X}_L^S$ denote the token features from the final student transformer block, and let $\mathbf{X}_{L-1}^T$ and $\mathbf{X}_L^T$ denote the token features from the last two teacher blocks. We attach a linear adapter $\phi(\cdot)$, implemented as a single learned linear layer applied token-wise, to map the student features to the teacher feature dimension. We then use the same adapted student representation to match both late teacher features:

$$\mathcal{L}_{\mathrm{distill}} = \sum_{l \in \{L-1, L\}} \left\| \phi\left(\mathbf{X}_L^S\right) - \mathbf{X}_l^T \right\|_2^2.$$

This objective is deliberately simple. It uses a single late student feature to align with multiple late teacher features, which encourages the compact student to absorb the teacher's high-level representation without introducing a heavy projection head. Using a minimal adapter keeps most of the representational burden on the student backbone itself, rather than allowing a high-capacity projection head to absorb the mismatch.

We study the effect of the aligned feature depth in the ablations and use the last two teacher blocks in the final recipe.

**Distillation setup.** Distillation is performed before downstream task adaptation. During this stage, the compact student is optimized only to match the detection-specialized teacher using $\mathcal{L}_{\text{distill}}$. The detailed training setups, including optimizer, data augmentation and training batch size, are provided in Section A.1. The smaller S detector is distilled from ECTeacher-S, while the larger M, L, and X detectors are distilled from ECTeacher-B, reflecting our finding that the teacher should be matched to the student capacity rather than made arbitrarily large. We also distill with both ImageNet-1K (Deng et al., 2009) and COCO (Lin et al., 2014) training set images, and further adapt the teacher on COCO detection before distillation, which makes the teacher more task-specific. The effects of aligned feature depth, teacher scale, optimizer choice, register tokens, distillation dataset, and teacher adaptation are all analyzed in the ablations. After distillation, the resulting backbone is reused for object detection. The backbone trained for object detection is then further transferred to instance segmentation and pose estimation. All downstream models are trained using only task-specific supervision from COCO (Lin et al., 2014).

### 3.4 Extensions to Human Pose Estimation and Instance Segmentation

ECPose and ECInsSeg reuse the distilled ECDet backbone and encoder, while adopting lightweight task-specific prediction modules. In all three tasks, training is performed only with task-specific COCO supervision (Lin et al., 2014).

#### 3.4.1 ECPose

**Pose decoder.** ECPose formulates human pose estimation as an end-to-end set prediction problem built on top of the distilled ECDet representation. The distilled backbone and encoder are kept unchanged, and only the detection head is replaced with a lightweight pose head. Following DETRPose (Janampa & Pattichis, 2025), the decoder maintains a fixed set of learned person queries, where each query is represented by one instance token and $K$ keypoint tokens corresponding to the $K$ human joints. This structured query design makes the pose layout explicit while preserving the same set-prediction interface as ECDet.

Each decoder layer updates these query tokens through self-attention and deformable cross-attention. Self-attention first exchanges information within each person query, allowing the instance token and keypoint tokens to reason jointly about a single pose. It also allows tokens of the same keypoint type across different queries to interact, which helps suppress duplicate predictions in crowded scenes. Deformable cross-attention then samples the encoded multi-scale features around the current reference points to refine the person and keypoint locations. The final decoder outputs are used to predict one person instance together with its keypoints.

**Training objective.** ECPose follows the same set-based matching paradigm as ECDet, with matching performed at the person-instance level. After matching, the training loss combines a Varifocal-style classification objective, a visible-joint regression objective, and an OKS loss:

$$\mathcal{L}_{\text{pose}} = \lambda_{\text{cls}}\mathcal{L}_{\text{cls}} + \lambda_{\text{kpt}}\mathcal{L}_{\text{kpt}} + \lambda_{\text{oks}}\mathcal{L}_{\text{oks}}.$$

The coefficient $\lambda_{\text{cls}}$ weights pose-quality-aware classification, while $\lambda_{\text{kpt}}$ and $\lambda_{\text{oks}}$ balance direct coordinate supervision and OKS-based pose quality. For the classification loss $\mathcal{L}_{\text{cls}}$, we use a Varifocal Loss-style objective (Zhang et al., 2021; Janampa & Pattichis, 2025), where the target score is defined by the standard COCO Object Keypoint Similarity (OKS) (Lin et al., 2014). Here, $\text{OKS}(i, j)$ measures the similarity between prediction $i$ and ground truth $j$ by normalizing the per-keypoint localization error with the person scale and keypoint-specific constants, and averaging over visible keypoints. For the keypoint regression loss $\mathcal{L}_{\text{kpt}}$, we use an $\ell_1$ loss between the predicted and ground-truth keypoint coordinates, applied only to visible joints:

$$\mathcal{L}_{\text{kpt}} = \sum_{k=1}^{K} v_{j,k} \left( |\hat{x}_{i,k} - x_{j,k}| + |\hat{y}_{i,k} - y_{j,k}| \right),$$

Table 2: **Performance comparison of real-time object detectors on COCO (Lin et al., 2014) val2017.** Best results without Objects365 pre-training are shown in **bold**. * denotes the actual training epochs, including additional epochs without augmentation. Latency is measured on an NVIDIA T4 GPU with batch size 1 under FP16 using TensorRT (v10.6), following RT-DETR (Lv et al., 2024) and D-FINE (Peng et al., 2025). YOLO latency includes NMS. Gray results indicate pre-training on Objects365 (Shao et al., 2019).

| Model | #Epochs | #Params. | GFLOPs | Latency (ms) | $AP^{val}$ | $AP_{50}^{val}$ | $AP_{75}^{val}$ | $AP_S^{val}$ | $AP_M^{val}$ | $AP_L^{val}$ |
|---|---|---|---|---|---|---|---|---|---|---|
| YOLOv9-S (Wang et al., 2024b) | 500 | 7M | 26 | 7.95 | 46.8 | 61.8 | 48.6 | 25.7 | 49.9 | 61.0 |
| YOLOv10-S (Wang et al., 2024a) | 500 | 7M | 22 | 2.52 | 46.3 | 63.0 | 50.4 | 26.8 | 51.0 | 63.8 |
| YOLO11-S (Glenn., 2024) | 500 | 9M | 22 | 7.05 | 46.6 | 63.4 | 50.3 | 28.7 | 51.3 | 64.1 |
| YOLOv12-S-turbo (Tian et al., 2025) | 600 | 9M | 19 | 8.14 | 47.6 | 64.5 | 51.5 | 28.3 | 52.7 | 65.9 |
| RT-DETRv2-S (Lv et al., 2024) | 120 | 20M | 60 | 4.61 | 48.1 | 65.1 | 52.1 | 30.2 | 51.5 | 63.9 |
| D-FINE-S (Peng et al., 2025) | 124* | 10M | 25 | 3.60 | 48.5 | 65.6 | 52.6 | 29.1 | 52.2 | 65.4 |
| DEIM-S (Huang et al., 2025b) | 132* | 10M | 25 | 3.60 | 49.0 | 65.9 | 53.1 | 30.4 | 52.6 | 65.7 |
| DEIMv2-S (Huang et al., 2025a) | 132* | 10M | 26 | 5.78 | 50.9 | 68.4 | 55.1 | 31.3 | 55.3 | 70.2 |
| RT-DETRv4-S (Liao et al., 2025) | 132* | 10M | 25 | 3.60 | 49.7 | 66.8 | 54.1 | 30.2 | 53.6 | 66.9 |
| LW-DETR-S (Chen et al., 2024) | 60 | 15M | 17 | 3.09 | 48.0 | 66.9 | 51.7 | 26.8 | 52.5 | 65.5 |
| D-FINE-S (Peng et al., 2025) | - | 10M | 25 | 3.60 | 50.7 | 67.6 | 55.1 | 32.7 | 54.6 | 66.5 |
| RF-DETR-S (Robinson et al., 2026) | - | 32M | 60 | 3.65 | 52.9 | 71.9 | 57.0 | 32.0 | 58.3 | 73.0 |
| YOLO26-S (Jocher & Qiu, 2026) | 70 | 10M | 21 | 2.59 | 47.8 | 64.6 | 52.1 | 29.1 | 52.5 | 64.3 |
| **ECDet-S** | 74 | 10M | 26 | 5.41 | **51.7** | **69.4** | **55.8** | **32.3** | **56.4** | **70.5** |
| YOLOv9-M (Wang et al., 2024b) | 500 | 20M | 76 | 10.05 | 51.4 | 67.2 | 54.6 | 32.0 | 55.7 | 66.4 |
| YOLOv10-M (Wang et al., 2024a) | 500 | 15M | 59 | 4.70 | 51.1 | 68.1 | 55.8 | 33.8 | 56.5 | 67.0 |
| YOLO11-M (Glenn., 2024) | 500 | 20M | 68 | 9.02 | 51.2 | 67.9 | 55.3 | 33.0 | 56.7 | 67.5 |
| YOLOv12-M-turbo (Tian et al., 2025) | 600 | 20M | 60 | 10.75 | 52.5 | 69.9 | 57.1 | 35.2 | 57.8 | 69.7 |
| RT-DETRv2-M (Lv et al., 2024) | 120 | 31M | 92 | 6.91 | 49.9 | 67.5 | 54.1 | 32.0 | 53.2 | 66.5 |
| D-FINE-M (Peng et al., 2025) | 124* | 19M | 57 | 5.66 | 52.3 | 69.8 | 56.4 | 33.2 | 56.5 | 70.2 |
| DEIM-M (Huang et al., 2025b) | 102* | 19M | 57 | 5.66 | 52.7 | 70.0 | 57.3 | 35.3 | 56.7 | 69.5 |
| DEIMv2-M (Huang et al., 2025a) | 102* | 18M | 52 | 8.80 | 53.0 | 70.2 | 57.4 | 34.2 | 57.4 | 71.5 |
| RT-DETRv4-M (Liao et al., 2025) | 102* | 19M | 57 | 5.66 | 53.5 | 71.1 | 58.1 | 34.9 | 57.7 | 72.1 |
| LW-DETR-M (Chen et al., 2024) | 60 | 28M | 43 | 5.27 | 52.6 | 69.9 | 56.7 | 32.6 | 57.7 | 70.7 |
| D-FINE-M (Peng et al., 2025) | - | 19M | 57 | 5.66 | 55.1 | 72.6 | 59.7 | 37.9 | 59.4 | 71.7 |
| RF-DETR-M (Robinson et al., 2026) | - | 34M | 79 | 4.62 | 54.7 | 73.5 | 59.2 | 36.1 | 59.7 | 73.8 |
| YOLO26-M (Jocher & Qiu, 2026) | 80 | 20M | 68 | 4.54 | 52.5 | 69.8 | 57.2 | 36.2 | 56.9 | 68.5 |
| **ECDet-M** | 62 | 18M | 53 | 7.98 | **54.3** | **72.2** | **58.7** | **35.9** | **59.1** | **72.7** |
| YOLOv9-C (Wang et al., 2024b) | 500 | 25M | 102 | 10.64 | 53.0 | 70.2 | 57.8 | 36.2 | 58.5 | 69.3 |
| YOLOv10-L (Wang et al., 2024a) | 500 | 24M | 120 | 7.38 | 53.2 | 70.1 | 58.1 | 35.8 | 58.5 | 69.4 |
| YOLO11-L (Glenn., 2024) | 500 | 25M | 87 | 10.34 | 53.4 | 70.1 | 58.2 | 35.6 | 59.1 | 69.2 |
| YOLOv12-L-turbo (Tian et al., 2025) | 600 | 27M | 82 | 14.39 | 53.8 | 71.0 | 58.6 | 36.9 | 59.4 | 71.0 |
| RT-DETRv2-L (Lv et al., 2024) | 72 | 42M | 136 | 9.29 | 53.4 | 71.6 | 57.4 | 36.1 | 57.9 | 70.8 |
| D-FINE-L (Peng et al., 2025) | 74* | 31M | 91 | 8.10 | 54.0 | 71.6 | 58.4 | 36.5 | 58.0 | 71.9 |
| DEIM-L (Huang et al., 2025b) | 58* | 31M | 91 | 8.10 | 54.7 | 72.4 | 59.4 | 36.9 | 59.6 | 71.8 |
| DEIMv2-L (Huang et al., 2025a) | 68* | 32M | 97 | 10.47 | 56.0 | 73.5 | 61.1 | 37.6 | 60.9 | 74.9 |
| RT-DETRv4-L (Liao et al., 2025) | 58* | 31M | 91 | 8.10 | 55.4 | 73.0 | 60.3 | 37.1 | 60.1 | 72.9 |
| LW-DETR-L (Chen et al., 2024) | 60 | 47M | 72 | 8.25 | 56.1 | 74.6 | 60.9 | 37.2 | 60.4 | 73.0 |
| D-FINE-L (Peng et al., 2025) | - | 31M | 91 | 8.10 | 57.1 | 74.7 | 62.0 | 40.0 | 61.5 | 74.2 |
| RF-DETR-L (Robinson et al., 2026) | - | 34M | 126 | 7.38 | 56.5 | 75.1 | 61.3 | 39.0 | 61.0 | 73.9 |
| YOLO26-L (Jocher & Qiu, 2026) | 60 | 25M | 86 | 6.20 | 54.3 | 71.5 | 59.4 | 37.8 | 58.6 | 70.3 |
| **ECDet-L** | 50 | 31M | 101 | 10.49 | **57.0** | **75.1** | **61.7** | **38.7** | **62.5** | **75.0** |
| YOLOv9-E (Wang et al., 2024b) | 500 | 57M | 189 | 19.68 | 55.6 | 72.8 | 60.6 | 40.2 | 61.0 | 71.4 |
| YOLOv10-X (Wang et al., 2024a) | 500 | 30M | 160 | 10.47 | 54.4 | 71.3 | 59.3 | 37.0 | 59.8 | 70.9 |
| YOLO11-X (Glenn., 2024) | 500 | 57M | 195 | 15.36 | 54.7 | 71.6 | 59.5 | 37.7 | 59.7 | 70.2 |
| YOLOv12-X-turbo (Tian et al., 2025) | 600 | 59M | 185 | 21.58 | 55.4 | 72.5 | 60.3 | 38.9 | 60.8 | 70.9 |
| RT-DETRv2-X (Lv et al., 2024) | 72 | 76M | 259 | 13.88 | 54.3 | 72.8 | 58.8 | 35.8 | 58.8 | 72.1 |
| D-FINE-X (Peng et al., 2025) | 74* | 62M | 202 | 12.90 | 55.8 | 73.7 | 60.2 | 37.3 | 60.5 | 73.4 |
| DEIM-X (Huang et al., 2025b) | 58* | 62M | 202 | 12.90 | 56.5 | 74.0 | 61.5 | 38.8 | 61.4 | 74.2 |
| DEIMv2-X (Huang et al., 2025a) | 58* | 50M | 152 | 13.75 | 57.8 | 75.3 | **63.2** | 39.1 | 62.9 | 75.9 |
| RT-DETRv4-X (Liao et al., 2025) | 58* | 62M | 202 | 12.90 | 57.0 | 74.6 | 62.1 | **39.5** | 61.9 | 74.8 |
| LW-DETR-X (Chen et al., 2024) | 60 | 118M | 174 | 16.06 | 58.3 | 76.9 | 63.3 | 40.9 | 63.3 | 74.8 |
| D-FINE-X (Peng et al., 2025) | - | 62M | 202 | 12.90 | 59.3 | 76.8 | 64.6 | 42.3 | 64.2 | 76.4 |
| RF-DETR-X (Robinson et al., 2026) | - | 126M | 300 | 14.79 | 58.6 | 77.4 | 63.8 | 40.3 | 63.9 | 76.2 |
| YOLO26-X (Jocher & Qiu, 2026) | 40 | 55M | 194 | 11.65 | 56.9 | 74.1 | 62.1 | 41.3 | 61.2 | 72.7 |
| **ECDet-X** | 50 | 49M | 151 | 12.70 | **57.9** | **76.0** | 62.9 | 38.7 | **63.4** | **76.1** |

where $v_{j,k}$ is the visibility indicator of the $k$-th ground-truth keypoint. Following DETRPose (Janampa & Pattichis, 2025), we also directly optimize the standard OKS loss: $\mathcal{L}_{\text{oks}} = 1 - \text{OKS}(i, j)$.

### 3.4.2 ECInsSeg

**Mask head.**  We extend ECDet to instance segmentation by attaching a lightweight query-based mask head, while keeping the backbone, encoder, and decoder unchanged. The same decoder queries used for box prediction are reused for mask prediction, so the task-specific change is confined to the mask branch. Following the general design used in MaskDINO (Li et al., 2023) and RF-DETR (Robinson et al., 2026), we build this branch from a higher-resolution encoded feature map.

Specifically, we use $\mathcal{F}^{(8)}$ as the input to the mask branch and resize it to the target mask resolution. The resized feature is processed by a depthwise convolution and a lightweight MLP to produce dense pixel embeddings, while each decoder query is projected into the same embedding space with a lightweight MLP. The mask logits for each query are then obtained by taking the dot product between its projected query embedding and the dense pixel embeddings at every spatial location. This design adds dense mask prediction with minimal extra parameters while preserving the shared ECDet representation.

**Training objective.**  Training follows the standard DETR assignment procedure, extended with mask supervision. The overall instance segmentation loss combines the detection losses with the mask losses:

$$\mathcal{L}_{\text{insseg}} = \mathcal{L}_{\text{det}} + \lambda_{\text{mask}}\mathcal{L}_{\text{mask}} + \lambda_{\text{dice}}\mathcal{L}_{\text{dice}},$$

The coefficient $\lambda_{\text{mask}}$ weights the dense mask supervision, while $\lambda_{\text{dice}}$ balances region-overlap quality relative to the inherited detection objective. For the mask term $\mathcal{L}_{\text{mask}}$, we use a pixel-wise mask classification loss applied only to positive matched queries. For the overlap term $\mathcal{L}_{\text{dice}}$, we use a standard Dice loss (Milletari et al., 2016) on the matched masks. The Hungarian matching procedure is extended to incorporate mask supervision in addition to the standard detection costs. Because the backbone and encoder are unchanged, ECInsSeg provides a direct test of whether the representation distilled for detection transfers to a second dense prediction task.

## 4 Experiments

We evaluate EdgeCrafter on three dense prediction tasks: object detection, human pose estimation, and instance segmentation. Unless otherwise specified, all models are trained and evaluated on COCO (Lin et al., 2014), and we report the standard task-specific metrics defined by the official evaluation protocol. Note that implementation details are provided in the Appendix A.2.

### 4.1 Main Results

**Object detection.**  Table 2 compares ECDet with recent efficient detectors on COCO and highlights a strong accuracy–efficiency trade-off across all model scales, despite training our models only with COCO supervision. In the small regime, ECDet-S reaches 51.7 AP with only 10M parameters and 26 GFLOPs, outperforming recent detectors in the same budget range such as RT-DETRv4-S (Liao et al., 2025) and DEIMv2-S (Huang et al., 2025a), while also exceeding several larger models. The same trend continues as the model scales up: ECDet-M reaches 54.3 AP with 18M parameters, ECDet-L improves to 57.0 AP with 31M parameters, and ECDet-X reaches 57.9 AP with 49M parameters and 151 GFLOPs. Notably, these larger variants remain competitive with, and in several cases outperform, baselines that benefit from additional Objects365 pretraining (Shao et al., 2019), indicating that task-specialized distillation can partially offset the need for heavier external supervision.

In terms of latency, ECDet is not always the absolute fastest model in every budget range, which is expected given the current software and hardware optimization gap for ViT-based architectures. However, the measured runtimes remain within a practical range for real-time or near-real-time deployment on the reported hardware, while the accuracy gains are substantial. This trade-off is still meaningful for real applications, since edge deployment is often constrained not only by runtime but also by available compute and storage budgets. For

Table 3: **Performance comparison of real-time pose estimation methods on COCO `val2017`, grouped by model scale.** Best results trained without Objects365 are highlighted in bold. The **Extra Sup.** column indicates whether a method uses additional supervision beyond COCO training data, where `O365` denotes Objects365 pretraining (Shao et al., 2019).

| Model | Extra Sup. | #Params. | GFLOPs | Latency (ms) | $AP^{val}$ | $AP_{50}^{val}$ | $AP_{75}^{val}$ | $AP_{M}^{val}$ | $AP_{L}^{val}$ | $AR^{val}$ |
|---|---|---|---|---|---|---|---|---|---|---|
| RTMO-S (Lu et al., 2024) | – | 9.9M | 30.7 | 3.78 | 67.7 | 87.8 | 73.7 | - | - | 71.5 |
| YOLO11-Pose-S (Glenn., 2024) | – | 9.9M | 23.2 | 4.54 | 58.9 | 86.3 | 64.8 | 54.0 | 68.0 | 66.1 |
| YOLO26-Pose-S (Jocher & Qiu, 2026) | – | 10.4M | 23.9 | 2.70 | 63.1 | 86.6 | 68.8 | 56.5 | 73.7 | 69.0 |
| DETRPose-S (Janampa & Pattichis, 2025) | O365 | 11.5M | 33.1 | 5.12 | 67.0 | 87.6 | 72.8 | 60.2 | 77.4 | 73.5 |
| **ECPose-S** | – | 9.9M | 30.4 | 5.54 | **68.9** | **89.1** | **75.2** | **60.7** | **81.1** | **74.6** |
| RTMO-M (Lu et al., 2024) | – | 22.6M | 69.0 | 6.21 | 70.9 | 89.0 | 77.8 | - | - | 74.7 |
| YOLO11-Pose-M (Glenn., 2024) | – | 20.9M | 71.7 | 6.65 | 64.9 | 89.4 | 72.4 | 62.2 | 71.6 | 72.2 |
| YOLO26-Pose-M (Jocher & Qiu, 2026) | – | 21.5M | 73.1 | 4.75 | 68.8 | 89.6 | 75.5 | 64.0 | 77.2 | 74.6 |
| DETRPose-M (Janampa & Pattichis, 2025) | O365 | 20.8M | 67.3 | 7.90 | 69.4 | 89.2 | 75.4 | 63.2 | 79.0 | 75.5 |
| **ECPose-M** | – | 19.8M | 62.8 | 9.25 | **72.4** | **90.9** | **78.6** | **65.2** | **83.6** | **78.2** |
| RTMO-L (Lu et al., 2024) | – | 44.8M | 136.7 | 8.35 | 72.4 | 89.9 | 78.8 | - | - | 76.8 |
| YOLO11-Pose-L (Glenn., 2024) | – | 26.2M | 90.7 | 7.95 | 66.1 | 89.9 | 73.6 | 63.2 | 73.1 | 73.3 |
| YOLO26-Pose-L (Jocher & Qiu, 2026) | – | 25.9M | 91.3 | 6.18 | 70.4 | 90.5 | 77.4 | 65.7 | 78.4 | 75.9 |
| DETRPose-L (Janampa & Pattichis, 2025) | O365 | 32.8M | 107.1 | 11.51 | 72.5 | 90.6 | 79.0 | 66.3 | 82.2 | 78.7 |
| **ECPose-L** | – | 34.3M | 111.7 | 11.83 | **73.5** | **91.7** | **79.9** | **66.4** | **84.4** | **78.8** |
| ED-Pose (Yang et al., 2023) | – | 218.0M | 422.6 | - | 74.3 | 91.5 | **81.7** | **68.5** | 82.7 | - |
| YOLO11-Pose-X (Glenn., 2024) | – | 58.8M | 203.3 | 12.95 | 69.5 | 91.1 | 77.4 | 66.6 | 76.0 | 76.3 |
| YOLO26-Pose-X (Jocher & Qiu, 2026) | – | 57.6M | 201.7 | 11.05 | 71.6 | 91.6 | 78.9 | 67.4 | 79.5 | 77.2 |
| DETRPose-X (Janampa & Pattichis, 2025) | O365 | 73.3M | 239.5 | 18.89 | 73.3 | 90.5 | 79.4 | 67.5 | 82.7 | 79.4 |
| **ECPose-X** | – | 50.6M | 172.2 | 14.31 | **74.8** | **92.2** | 81.5 | 68.0 | **85.4** | **80.1** |

Table 4: **Performance comparison of real-time instance segmentation models on COCO (Lin et al., 2014) `val2017`, grouped by model scale.** Best results trained without Objects365 are highlighted in bold. The **Extra Sup.** column indicates whether a method uses additional supervision beyond COCO, where `O365+SAM2` denotes Objects365 pretraining (Shao et al., 2019) with SAM2 (Ravi et al., 2025) pseudo-instance masks before COCO fine-tuning.

| Model | Extra Sup. | #Params. | GFLOPs | Latency (ms) | $AP^{val}$ | $AP_{50}^{val}$ | $AP_{75}^{val}$ | $AP_{S}^{val}$ | $AP_{M}^{val}$ | $AP_{L}^{val}$ |
|---|---|---|---|---|---|---|---|---|---|---|
| YOLOv8-Seg-S (Glenn., 2023) | – | 11.8M | 42.6 | 7.08 | 36.8 | 58.1 | 38.9 | 17.3 | 41.3 | 53.7 |
| YOLO11-Seg-S (Glenn., 2024) | – | 10.1M | 35.5 | 7.20 | 37.8 | 59.9 | 40.4 | 19.8 | 42.7 | 55.2 |
| YOLO26-Seg-S (Jocher & Qiu, 2026) | O365+SAM2 | 10.4M | 34.2 | 3.51 | 40.0 | 61.5 | 43.0 | 21.0 | 44.5 | 57.3 |
| RF-DETR-Seg-S (Robinson et al., 2026) | O365+SAM2 | 33.7M | 70.6 | 4.81 | 43.1 | 66.2 | 45.9 | 21.9 | 48.5 | 64.1 |
| **ECInsSeg-S** | – | 10.3M | 33.1 | 6.96 | **43.0** | **65.7** | **46.0** | **20.8** | **46.3** | **65.9** |
| YOLOv8-Seg-M (Glenn., 2023) | – | 27.3M | 110.2 | 9.61 | 40.8 | 63.6 | 43.8 | 22.0 | 46.0 | 58.2 |
| YOLO11-Seg-M (Glenn., 2024) | – | 22.4M | 113.2 | 9.18 | 41.5 | 65.1 | 44.5 | **23.0** | 47.2 | 59.1 |
| YOLO26-Seg-M (Jocher & Qiu, 2026) | O365+SAM2 | 23.6M | 121.5 | 6.53 | 44.1 | 66.8 | 47.7 | 25.6 | 48.9 | 60.2 |
| RF-DETR-Seg-M (Robinson et al., 2026) | O365+SAM2 | 35.7M | 102.0 | 6.35 | 45.3 | 68.4 | 48.8 | 25.5 | 50.4 | 65.3 |
| **ECInsSeg-M** | – | 20.1M | 64.2 | 9.85 | **45.2** | **68.2** | **48.3** | 22.9 | **49.0** | **68.1** |
| YOLOv8-Seg-L (Glenn., 2023) | – | 46.0M | 220.5 | 12.29 | 42.6 | 66.2 | 45.5 | 23.9 | 47.9 | 59.5 |
| YOLO11-Seg-L (Glenn., 2024) | – | 27.6M | 132.2 | 10.55 | 42.9 | 67.0 | 46.2 | 24.6 | 48.7 | 60.6 |
| YOLO26-Seg-L (Jocher & Qiu, 2026) | O365+SAM2 | 28.0M | 139.8 | 7.77 | 45.5 | 68.7 | 49.2 | 27.1 | 50.4 | 62.8 |
| RF-DETR-Seg-L (Robinson et al., 2026) | O365+SAM2 | 36.2M | 151.1 | 9.42 | 47.1 | 70.5 | 50.9 | 28.4 | 52.1 | 65.6 |
| **ECInsSeg-L** | – | 33.6M | 110.8 | 12.56 | **47.1** | **70.9** | **50.5** | **24.8** | **51.1** | **69.6** |
| MaskDINO (Li et al., 2023) | – | 52.1M | 286 | - | 46.3 | 69.0 | 50.7 | 26.1 | 49.3 | 66.1 |
| YOLOv8-Seg-X (Glenn., 2023) | – | 71.8M | 344.1 | 16.20 | 43.4 | 67.1 | 46.5 | 25.6 | 48.9 | 60.4 |
| YOLO11-Seg-X (Glenn., 2024) | – | 62.1M | 296.4 | 15.54 | 43.8 | 68.5 | 47.1 | 25.7 | 49.7 | 61.4 |
| YOLO26-Seg-X (Jocher & Qiu, 2026) | O365+SAM2 | 62.8M | 313.5 | 15.13 | 47.0 | 70.8 | 51.1 | 29.7 | 51.8 | 63.1 |
| RF-DETR-Seg-X (Robinson et al., 2026) | O365+SAM2 | 38.1M | 260.0 | 15.42 | 48.8 | 72.2 | 53.1 | 30.6 | 53.3 | 65.9 |
| **ECInsSeg-X** | – | 49.9M | 168.1 | 14.96 | **48.4** | **72.2** | **52.0** | **26.3** | **52.7** | **71.1** |

example, commercial SoCs such as Rockchip RK3568 integrate only a 1 TOPS NPU for INT8 inference, making compact models attractive not only for throughput but also for reducing flash and memory usage (Rockchip, 2022). Overall, the results show that the distilled compact ViT backbone scales favorably for edge object detection without relying on large auxiliary pretraining pipelines or substantially larger detectors.

**Transfer to human pose estimation.** Table 3 shows that the detection-distilled representation transfers effectively to human pose estimation with only task-specific decoder changes and COCO keypoint supervision. In the small-scale regime, ECPose-S reaches 68.9 AP with only 9.9M parameters, significantly outperforming YOLO11-Pose-S (Glenn., 2024) (58.9 AP) and YOLO26-Pose-S (Jocher & Qiu, 2026) (63.1 AP), while also exceeding the Objects365-pretrained (Shao et al., 2019) DETRPose-S (Janampa & Pattichis, 2025) (67.0 AP). This trend continues at medium scale, where ECPose-M achieves 72.4 AP, not only surpassing its direct competitors but also matching the performance of much larger models such as RTMO-L (Lu et al., 2024) and DETRPose-L. At larger scales, ECPose-L and ECPose-X reach 73.5 AP and 74.8 AP, respectively, consistently outperforming the corresponding DETRPose models. Notably, ECPose-X even surpasses the heavy-weight ED-Pose (Yang et al., 2023) (74.3 AP) despite the latter having a massive 218M parameter count. The latency numbers again show that the proposed models are practical, even if they are not always the lowest-latency option in every scale regime, and the improved accuracy is obtained without expanding to substantially larger backbones. Taken together, these results indicate that the representation learned through detection-centered distillation is not limited to bounding-box prediction: it also transfers effectively to fine-grained keypoint localization while preserving a strong accuracy–efficiency trade-off across the full model family.

**Transfer to instance segmentation.** Table 4 evaluates whether the same detection-distilled representation can support dense mask prediction with a lightweight segmentation head trained only on COCO instance annotations. In the small regime, ECInsSeg-S reaches 43.0 AP with only 10.3M parameters, clearly outperforming YOLO11-Seg-S (Glenn., 2024) and nearly matching RF-DETR-Seg-S (Robinson et al., 2026) even though the latter is much larger and uses additional Objects365 pretraining (Shao et al., 2019) together with pseudo masks from SAM2 (Ravi et al., 2025). The same behavior remains stable at higher capacities: ECInsSeg-M reaches 45.2 AP and stays very close to RF-DETR-Seg-M with a smaller model and lower compute, while ECInsSeg-L matches RF-DETR-Seg-L at 47.1 AP with a lower FLOP budget. At the extra-large scale, ECInsSeg-X reaches 48.4 AP with 168.1 GFLOPs, approaching RF-DETR-Seg-X while remaining substantially more compute-efficient. Similar to detection and pose, the reported latency is practical for deployment even though ECInsSeg is not always the fastest entry in the table, and the compact parameter count remains important for memory- and storage-constrained devices. Overall, these results show that the distilled ECDet representation transfers beyond detection to instance segmentation without requiring a separate backbone redesign, while preserving a favorable mask-accuracy, parameter, and FLOP trade-off across the full scale range.

## 4.2 Analysis

### 4.2.1 Distillation Analysis

We first analyze the distillation recipe, since the paper's main claim is that compact ViTs become competitive when the teacher, data, and optimization strategy are chosen to match dense prediction. Unless otherwise specified, all ablations distill ECViT-T+ and evaluate the resulting backbone through the downstream performance of ECDet-M on COCO (Lin et al., 2014). We use downstream detection AP rather than the distillation loss as the evaluation metric, because lower feature-matching loss does not always translate into better dense-prediction performance.

**Teacher design and distillation data.** Table 5 summarizes the most important design choices in the distillation pipeline. First, teacher capacity should be matched to the student rather than increased without bound: DINOv3-B (Siméoni et al., 2025) gives the best result, while the smaller DINOv3-S is weaker and the larger DINOv3-L degrades performance despite its higher capacity. This suggests that an excessively strong teacher can create a representation gap that the compact student cannot effectively absorb. Second, adapting the teacher to detection on COCO before distillation improves AP from 53.5 to 54.3, confirming that teacher specificity matters for dense prediction. Third, adding COCO images to the distillation corpus improves AP from 54.1 to 54.3 over using ImageNet-1K alone (Deng et al., 2009), indicating that additional task-relevant images might provide complementary supervision beyond generic classification data. Taken together, these

Table 5: **Unified ablation on teacher design and distillation data.** All experiments use ECViT-T+ as the student. The teacher is initialized from DINOv3 (Siméoni et al., 2025), and "COCO pre-training" indicates that the teacher is first adapted as the ECDet backbone on COCO before distillation.

| Teacher Arch. | COCO Pre-training | Distillation Dataset | AP (%) |
|---|---|---|---|
| *Teacher capacity* | | | |
| DINOv3-S | ✓ | IN-1K + COCO | 54.0 |
| **DINOv3-B** | ✓ | IN-1K + COCO | **54.3** |
| DINOv3-L | ✓ | IN-1K + COCO | 52.6 |
| *Teacher task adaptation* | | | |
| DINOv3-B | ✗ | IN-1K + COCO | 53.5 |
| *Distillation dataset* | | | |
| DINOv3-B | ✓ | IN-1K | 54.1 |

Table 6: **Ablation on feature-alignment depth.** We vary how many of the teacher's final layers supervise the student and report downstream detection AP after distillation.

| Student | Student Layers | Teacher Layers | AP |
|---|---|---|---|
| ECViT-T+ | 1 | 1 | 53.6 |
| | 1 | 2 | 54.3 |
| | 1 | 3 | **54.6** |
| | 1 | 4 | 54.1 |
| | 2 | 2 | 54.2 |
| ECViT-S | 1 | 2 | **57.0** |
| | 1 | 3 | 56.9 |
| ECViT-S+ | 1 | 2 | **57.9** |
| | 1 | 3 | 57.5 |

Table 7: **Ablation on optimization and register tokens.** Unless otherwise specified, the default setting uses LARS (You et al., 2017) and one register token (Darcet et al., 2023).

| Optimizer | Register Number | AP |
|---|---|---|
| **Effect of optimizer** | | |
| AdamW | 1 | 54.0 |
| LARS | 1 | **54.3** |
| **Effect of register tokens** | | |
| LARS | 0 | 53.8 |
| LARS | 2 | 54.2 |
| LARS | 4 | 54.2 |

results support the final recipe used in EdgeCrafter: a detection-specialized, scale-matched teacher distilled on both ImageNet-1K and COCO.

**Feature alignment depth.** Table 6 studies how many late teacher features should supervise the student. Pure one-to-one alignment between the last teacher and student layers is clearly suboptimal, suggesting that a single-point matching objective does not transfer enough high-level supervision. Aligning the student's final layer to the teacher's last two layers improves performance consistently across all tested scales, and this is the configuration adopted in the final model. Although using the last three teacher layers gives a slight gain for ECViT-T+, that advantage does not carry over to the larger ECViT-S and ECViT-S+ variants. We therefore choose the last-two-layer alignment because it is the most stable option across scales, even if the very best result for one particular student is obtained with three layers.

**Optimization and register tokens.** Table 7 shows that the optimization recipe and the use of register tokens both matter for the final distilled backbone. Replacing AdamW (Loshchilov & Hutter, 2019) with LARS (You et al., 2017) improves AP from 54.0 to 54.3, indicating that LARS provides a better optimization regime for matching compact student features to a stronger teacher in this setting. The same table also shows that register tokens (Darcet et al., 2023) are beneficial: removing them lowers AP to 53.8, while adding one register restores performance to the best result. Increasing the number of registers beyond one does not produce further gains, with both two and four registers reaching 54.2 AP. We therefore adopt LARS together with a single register token as the default configuration, since this combination gives the best accuracy while keeping the student backbone compact.

Table 8: **Effect of patch embedding design and stem receptive field. Vanilla** uses a single $16 \times 16$ stride-16 convolution. **ConvStem** follows the convolutional-stem design used in the final model and consists of four $3 \times 3$ stride-2 convolutions (Xiao et al., 2021). Here, $d$ denotes the dilation rate of the final stem convolution, which controls the effective receptive field of the patch embedding stage. All results are reported on ECDet-M.

| Patch Embedding | Params (M) | FLOPs (G) | Latency (ms) | AP | $AP_S$ | $AP_M$ | $AP_L$ |
|---|---|---|---|---|---|---|---|
| Vanilla | 19.0 | 50.7 | 7.87 | 53.5 | 33.7 | 58.2 | 72.8 |
| **ConvStem** ($d$=1) | **19.2** | **53.1** | **7.98** | **54.3** | **35.9** | **59.1** | **72.7** |
| ConvStem ($d$=2) | 19.2 | 53.1 | - | 53.0 | 33.7 | 57.9 | 72.5 |
| ConvStem ($d$=3) | 19.2 | 53.1 | - | 53.6 | 33.2 | 58.6 | 72.6 |

Table 9: **Effect of layer fusion strategy.** We compare simple mean fusion over different late-layer ranges with concatenation and the STA fusion module from DEIMv2 (Huang et al., 2025a). The gray row marks the adopted default setting in ECDet-M.

| Fusion Method | Fused Layers | Params (M) | FLOPs (G) | Latency (ms) | AP |
|---|---|---|---|---|---|
| Mean | $L_{11}$ | 19.2 | 53.1 | 7.98 | 54.1 |
| Mean | $L_{10 \to 11}$ | 19.2 | 53.1 | 7.98 | **54.3** |
| Mean | $L_{9 \to 11}$ | 19.2 | 53.1 | 7.98 | **54.3** |
| Mean | $L_{6 \to 11}$ | 19.2 | 53.1 | 8.11 | 54.1 |
| Concat | $L_{10 \to 11}$ | 19.5 | 55.2 | 8.05 | 54.2 |
| STA (Huang et al., 2025a) | $L_5, L_7, L_{11}$ | 19.4 | 54.5 | 8.48 | **54.3** |

### 4.2.2 Detection Analysis

**Ablation study on patch embedding.** We evaluate the patch embedding design in a unified ablation that compares the vanilla large-stride embedding with the proposed convolutional stem, and further varies the dilation rate of the final stem convolution to control the receptive field. As shown in Table 8, the convolutional stem slightly increases computation over the vanilla embedding but yields a clear AP gain, especially on small objects. Increasing the dilation rate beyond the default setting degrades performance, particularly on small and medium objects, indicating that a progressively enlarged but still moderate receptive field is more suitable for dense localization.

**Ablation study on fusion methods.** Table 9 compares different feature-fusion strategies across the final transformer blocks. Averaging the last two layers offers the best accuracy–efficiency trade-off, while concatenation and the heavier STA fusion used in DEIMv2 (Huang et al., 2025a) provide no consistent advantage once parameter and FLOP costs are considered.

## 5 Conclusion

We presented EdgeCrafter, a unified compact ViT framework for edge dense prediction centered on ECDet. The key idea is to adapt a large pretrained ViT into a detection-specialized teacher and distill its representation into compact student backbones designed for edge deployment. Combined with a lightweight convolutional stem and simple multi-scale feature generation, this yields strong accuracy–efficiency trade-offs for object detection under compact parameter and FLOP budgets. The resulting detection-distilled representation also transfers effectively to instance segmentation and human pose estimation through lightweight task-specific prediction modules. Across these tasks, EdgeCrafter achieves strong results on COCO while remaining competitive with, and in several settings outperforming, methods that rely on additional external supervision. Overall, our results indicate that compact ViTs can serve as a practical foundation for edge dense prediction when paired with task-specialized distillation and edge-oriented architectural design.

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

# A  Appendix

## A.1  Distillation Pre-training.

Before downstream task training, we distill the compact ECViT backbones from detection-specialized teachers. Each teacher is constructed by adapting a pretrained DINOv3 model to object detection using the same formulation as ECDet. Specifically, we fine-tune DINOv3-B within the ECDet-X framework to obtain ECTeacher-B, and adapt DINOv3-S with ECDet-L to produce ECTeacher-S.

Distillation is then performed on a combined dataset comprising ImageNet-1K (Deng et al., 2009) and the COCO training split (Lin et al., 2014). All student backbones are trained for 50 epochs with LARS (You et al., 2017), using a linear warm-up for the first 5 epochs followed by cosine decay to $10^{-3}$ of the peak learning rate. Adopting the scaling rule from Lightly (2026), the peak learning rate is determined by base learning rate $\cdot \sqrt{B/1536}$, where $B$ denotes the total batch size. We set the base learnling rate to 4.0 for ECViT-T/T+ and 9.0 for ECViT-S/S+, respectively. We employ a weight decay of $10^{-6}$ and distribute the training across $2\times$ NVIDIA RTX 4090 GPUs with a total batch size of 128. Distillation uses random resized cropping at $224 \times 224$, horizontal flipping, color jitter, grayscale conversion, Gaussian blur, and Mixup augmentation (Zhang et al., 2018).

## A.2  Implementation Details for ECDet ECPose and ECInsSeg

### A.2.1  Object Detection

We provide detailed hyper-parameters in Table 10. Precisely, for object detection, we train ECDet on the COCO `train2017` split and evaluate on `val2017`. All detector variants use an input resolution of $640 \times 640$. We report the standard COCO detection metrics, including AP, $AP_{50}$, $AP_{75}$, $AP_S$, $AP_M$, and $AP_L$. We train the ECDet variants using the AdamW optimizer with a total batch size of 32 on $4\times$ NVIDIA RTX 4090 GPUs. The Base LR is fixed at $5 \times 10^{-4}$, while the Backbone LR is specifically scaled down from $2.5 \times 10^{-5}$ to $2.5 \times 10^{-6}$ as model capacity increases to preserve distilled representations. We adopt a progressive training schedule, ranging from 74 epochs (S) to 50 epochs (L/X), all concluding with a 2-epoch fine-tuning stage that disables heavy augmentations. For data augmentation, we employ Mosaic and Mixup with a probability of 0.75 for smaller variants and 1.0 for larger ones, active during the first half of the training duration to facilitate robust feature learning. The weight decay is set to $10^{-4}$, slightly adjusted to $1.25 \times 10^{-4}$ for the L and X variants.

**About latency.**  Latency measurement is a non-trivial problem, as it is highly affected by external GPU conditions, particularly thermal behavior and cooling efficiency. Even under identical hardware, variations in GPU temperature can lead to noticeable differences in inference speed. For example, in our X-scale model, a change in GPU idle temperature from 29 °C to 35 °C results in an approximately 0.2 ms difference in measured latency. As a result, latency numbers reported by different users or systems may not be directly comparable. To ensure a fair and controlled evaluation, we measure the latency of all models under the same experimental conditions. Specifically, all experiments are conducted on an NVIDIA T4 GPU with the idle temperature stabilized at 29 °C. We report inference latency with batch size 1 under FP16 precision using TensorRT (v10.6), following the evaluation protocol of D-FINE (Peng et al., 2025) and RT-DETR (Lv et al., 2024).

### A.2.2  Human Pose Estimation

The training protocol for human pose estimation largely follows the configuration of ECDet and DETR-Pose (Janampa & Pattichis, 2025), with several task-specific adjustments summarized in Table 11. Specifically, we increase the Total Batch Size to 64. The training duration is extended to 92 epochs for smaller variants (S/M) and 74 epochs for larger variants (L/X), maintaining a 2-epoch final fine-tuning stage without augmentations. For the optimization objective, we employ a combination of classification, keypoint regression, and OKS losses, with the loss weights set to $\lambda_{\text{cls}} = 2$, $\lambda_{\text{kpt}} = 10$, and $\lambda_{\text{oks}} = 4$, respectively. The data

Table 10: **Hyper-parameters for ECDet.**

| Setting | ECDet-S | ECDet-M | ECDet-L | ECDet-X |
|---|---|---|---|---|
| Optimizer | AdamW | AdamW | AdamW | AdamW |
| Backbone LR | 2.5e-5 | 2.5e-5 | 5e-6 | 2.5e-6 |
| Base LR | 5e-4 | 5e-4 | 5e-4 | 5e-4 |
| LR decay | 0.5 | 0.5 | 0.5 | 0.5 |
| Weight Decay | 1e-4 | 1e-4 | 1.25e-4 | 1.25e-4 |
| Epochs (w/ + w/o Aug.) | 72 + 2 | 60 + 2 | 48 + 2 | 48 + 2 |
| $\text{Prob}_{\text{mosaic}}$ | 0.75 | 0.75 | 1 | 1 |
| $\text{Epochs}_{\text{mosaic}}$ | 36 | 30 | 24 | 24 |
| $\text{Prob}_{\text{mixup}}$ | 0.75 | 0.75 | 1 | 1 |
| $\text{Epochs}_{\text{mixup}}$ | 36 | 30 | 24 | 24 |
| Total Batch Size | 32 | 32 | 32 | 32 |
| $\lambda_{\text{cls}}$ | 1 | 1 | 1 | 1 |
| $\lambda_{\ell_1}$ | 5 | 5 | 5 | 5 |
| $\lambda_{\text{giou}}$ | 2 | 2 | 2 | 2 |
| $\lambda_{\text{ddf}}$ | 1.5 | 1.5 | 1.5 | 1.5 |
| $\lambda_{\text{fgl}}$ | 0.15 | 0.15 | 0.15 | 0.15 |

Table 11: **Hyper-parameters for ECPose.**

| Setting | ECPose-S | ECPose-M | ECPose-L | ECPose-X |
|---|---|---|---|---|
| Optimizer | AdamW | AdamW | AdamW | AdamW |
| Backbone LR | 2.5e-5 | 2.5e-5 | 2.5e-6 | 2.5e-6 |
| Base LR | 5e-4 | 5e-4 | 5e-4 | 5e-4 |
| Weight Decay | 1e-4 | 1e-4 | 1.25e-4 | 1.25e-4 |
| Epochs (w/ + w/o Aug.) | 90 + 2 | 90 + 2 | 72 + 2 | 72 + 2 |
| $\text{Prob}_{\text{mosaic}}$ | 0.5 | 0.5 | 0.5 | 0.5 |
| $\text{Epochs}_{\text{mosaic}}$ | 45 | 45 | 48 | 48 |
| $\text{Prob}_{\text{mixup}}$ | 0.5 | 0.5 | 0.5 | 0.5 |
| $\text{Epochs}_{\text{mixup}}$ | 45 | 45 | 48 | 48 |
| $\text{Prob}_{\text{copypaste}}$ | 0.5 | 0.5 | 0.5 | 0.5 |
| $\text{Epochs}_{\text{copypaste}}$ | 45 | 45 | 48 | 48 |
| Total Batch Size | 64 | 64 | 64 | 64 |
| $\lambda_{\text{cls}}$ | 2 | 2 | 2 | 2 |
| $\lambda_{\text{kpt}}$ | 10 | 10 | 10 | 10 |
| $\lambda_{\text{oks}}$ | 4 | 4 | 4 | 4 |

augmentation strategy remains consistent with detection, and we employ a moderated Mosaic and Mixup probability of 0.5.

### A.2.3 Instance Segmentation

We adopt the same training protocols as ECDet, including the optimizer, learning rate schedules, and data augmentations, while keeping the total batch size consistent across 8 GPUs. As summarized in Table 12, we incorporate a binary cross-entropy mask loss $\mathcal{L}_{\text{mask}}$ and a Dice loss $\mathcal{L}_{\text{dice}}$ (both weighted by $\lambda = 5.0$) to supervise mask prediction. To balance the multi-task learning objectives across the mask and box branches, the classification weight $\lambda_{\text{cls}}$ is increased to 2.0, while the box regression weights ($\lambda_{\ell_1}$, $\lambda_{\text{giou}}$) are moderated to 1.0 relative to the original detection task.

Table 12: **Hyper-parameters for ECInsSeg.**

| Setting | ECInsSeg-S | ECInsSeg-M | ECInsSeg-L | ECInsSeg-X |
|---|---|---|---|---|
| $\lambda_{\mathrm{cls}}$ | 2 | 2 | 2 | 2 |
| $\lambda_{\ell_1}$ | 1 | 1 | 1 | 1 |
| $\lambda_{\mathrm{giou}}$ | 1 | 1 | 1 | 1 |
| $\lambda_{\mathrm{ddf}}$ | 1.5 | 1.5 | 1.5 | 1.5 |
| $\lambda_{\mathrm{fgl}}$ | 0.15 | 0.15 | 0.15 | 0.15 |
| $\lambda_{\mathrm{mask}}$ | 5 | 5 | 5 | 5 |
| $\lambda_{\mathrm{dice}}$ | 5 | 5 | 5 | 5 |

### A.3   Training Cost

Table 13 further reports the training cost in GPU hours to provide a more comprehensive view of efficiency beyond the mere number of epochs. Compared to RT-DETRv4-M, ECDet-M achieves a performance gain of **+0.8 AP** while reducing the training time by approximately **49%** ($\sim$96 vs. $\sim$190 GPU hours).

Table 13: **Comparison of training cost in GPU hours on COCO.** All models are trained on **4×
NVIDIA RTX 4090 GPUs**. Gray results indicates pre-training on Objects365 (Shao et al., 2019) followed by fine-tuning on COCO.

| Method | #Epochs | GPU Hours ↓ | AP |
|---|---|---|---|
| YOLO26-M | 80 | $\sim$24 | 52.5 |
| YOLO11-M | 500 | $\sim$133 | 51.2 |
| RT-DETRv4-M | 102 | $\sim$190 | 53.5 |
| ECDet-M | 74 | **$\sim$96** | **54.3** |

