# OpenReview forum: "EdgeCrafter: Compact ViTs for Edge Dense Prediction via Task-Specialized Distillation"
_TMLR — Under review for TMLR_

### Review · Reviewer_zrgF · 2026-04-26

**Summary Of Contributions:**

This paper proposes ECDet, a family of ViT-based dense prediction models that outperforms baselines under low parameter counts. The paper contributes to both model architecture design and training method design. From the model side, lightweight convolutional patch embedding and multiscale feature projections are proposed to learn task-specific features with light parameters. Form the training perspective, distillation and finegrained search on hyperparameter are conducted to boost the performance.

### Key Strength
1. This paper is very easy to read. The motivation is clear, discussion is highly detailed, and the structure is well organized.
2. The paper provides novel design in patch embedding and multi-scale feature extractor specially tailored for lightweight dense prediction.
3. Solid experimental results are provided to show the improvement of the proposed method over baseline.

### Key weakness
1. Though not specifically discussed by the author, it appears that the training cost, both in terms of the amount of data utilized and in GPU hours may largely exceed that of baseline due to the pretraining-distillation framework. A discussion of training cost is needed to fiarly judge the method.
2. Previous work investigating the performance gain of DeiT over ResNet [1] has observed that performance improvement of new models over baseline can come from the improvement of the training pipeline (e.g. dataset, augmentation, optimizer, etc.), rather than the new model design. To this end, it would be interesting to see how Yolo and DETR baseline will improve if trained following the same distillation procedure as proposed and using same optimizer and hyperparameters. This will justify if the improvement mainly comes from distillation or from the new architecture design.

[1] Wightman, Ross, Hugo Touvron, and Hervé Jégou. "Resnet strikes back: An improved training procedure in timm." arXiv preprint arXiv:2110.00476 (2021).

**Audience:**

Yes

**Audience Explanation:**

The paper proposes new model for the dense prediction vision task. The research domain is highly active with a large amount of audience.

**Broader Impact Concerns:**

No broader impact concerns.

**Claims And Evidence:**

Yes

**Claims Explanation:**

Detailed experimental results are provided showcasing the performance gain achieved by the proposed method over baseline. Training details are also clearly presented, indicating good reproducability.

**Requested Changes:**

1. Please include discussion on the overall training cost of the full training pipeline, including the amount of data used and total GPU hours.
2. Please try to train baseline models, like YOLO, with the same training data, distillation teacher, and hyperparameter choices.

---

> ### Author Response · Authors · 2026-07-07
>
> We sincerely thank Reviewer zrgF for the careful reading of our paper and the positive assessment. We appreciate the recognition of the paper's clarity, technical contributions, and experimental results. Below, we respond to each concern in detail.
>
> >  Please include discussion on the overall training cost of the full training pipeline, including the amount of data used and total GPU hours.
>
> The overall training cost of our framework is summarized below.
>
> |   Method   |        Pre-training Dataset         |   Pre-training Images   |   Pre-training GPU Hours   |   COCO Detection Epochs   |   Detection GPU Hours   |   COCO AP   |
> |:----------:|:-----------------------------------:|:-----------------------:|:--------------------------:|:-------------------------:|:-----------------------:|:-----------:|
> | YOLO11-M | - | - | - | 600 | $\sim$160 | 51.2 |
> | RT-DETRv4-M | ImageNet-22K | $\sim$14.20M | - | 102 | $\sim$190 | 53.5 |
> | **ECDet-M** | ImageNet-1K train + COCO train2017 | $\sim$1.28M + $\sim$0.12M | $\sim$34 | 74 | $\sim$96 | **54.3** |
>
> Taking ECDet-M as an example, the complete training pipeline consists of two stages.
>
> Stage 1: Backbone Distillation. We distill the compact ECViT backbone using unlabeled images from ImageNet-1K train and COCO train2017, corresponding to approximately 1.28M ImageNet images and 0.12M COCO images. Importantly, no ImageNet labels or COCO annotations are used during this stage. **The distillation is performed on 2 RTX 4090 GPUs for approximately 17 hours, corresponding to 34 GPU hours.**
>
> Stage 2: Object Detection Training. We then train the detector on COCO train2017 using detection annotations. **This stage is conducted on 4 RTX 4090 GPUs for approximately 24 hours, corresponding to 96 GPU hours.
>
> **Therefore, the total training cost of ECDet-M is approximately 130 GPU hours.**
>
> For comparison:
> - YOLO11-M requires approximately 160 GPU hours for COCO detection training and achieves 51.2 AP.
> - RT-DETRv4-M requires approximately 190 GPU hours for COCO detection training and achieves 53.5 AP.
> - ECDet-M requires only 96 GPU hours for the COCO detection stage and achieves 54.3 AP. Even after including the additional 34 GPU hours for backbone distillation, the total training cost remains comparable to these real-time detectors while achieving higher detection accuracy.
>
> **All GPU-hour statistics are measured using the same RTX 4090 GPUs for fair comparison.**
>
> We also note that RT-DETRv4 adopts [HGNet](https://github.com/PaddlePaddle/PaddleClas/blob/f1233c18455b8acde4fc42ab0bea575fa06daa8e/docs/en/models/PP-HGNetV2_en.md) as its backbone, which is reported to be self-supervised pre-trained on ImageNet-22K, containing approximately 14.20M images. However, to the best of our knowledge, the GPU cost of this ImageNet-22K pre-training stage is not reported in either the paper or the official repository. Therefore, our table explicitly separates the backbone pre-training/distillation stage from the downstream COCO detection training stage for a clearer comparison.
>
> **Importantly, the backbone distillation stage is a one-time cost. Once the ECViT backbone has been distilled, the same representation is reused across object detection, instance segmentation, and human pose estimation.** Consequently, when multiple dense prediction tasks are considered, the distillation cost can be amortized across ECDet, ECInsSeg, and ECPose, rather than being incurred independently for each task.

---

> > ### Author Response · Authors · 2026-07-07
> >
> > > Please try to train baseline models, like YOLO, with the same training data, distillation teacher, and hyperparameter choices.
> >
> > We thank the reviewer for this valuable suggestion. Following the reviewer's recommendation, we trained a YOLO11-M baseline using the same distillation data, teacher, and training schedule as ECDet-M. Specifically, we used ImageNet-1K train + COCO train2017 images for distillation, adopted the same ECDet teacher, and applied the same 50 epoch distillation schedule.
> >
> > Since the final feature map of YOLO11-M has a stride of 32, whereas the teacher feature map has a stride of 16, we first bilinearly upsampled the YOLO feature map to match the teacher's spatial resolution before performing feature alignment. After distillation, the detector was trained on COCO for 600 epochs, following the standard YOLO11-M training protocol to ensure a fair comparison.
> >
> > The results are summarized below.
> >
> > | Model | Distillation Data | Distillation Epochs | COCO Detection Epochs | AP |
> > |-------|-------------------|--------------------:|----------------------:|---:|
> > | YOLO11-M | - | - | 600 | 51.6 |
> > | YOLO11-M + Our Distillation | ImageNet-1K train + COCO train2017 | 50 | 600 | **51.7** |
> >
> > We observe only a modest improvement of **+0.1 AP**, indicating that directly applying our feature-level distillation strategy to YOLO does not yield the same benefit as it does for ECDet.
> >
> > We believe that the primary reason is the architectural mismatch between the teacher and the student. Both the teacher and ECDet are ViT-based models, making their spatial representations and prediction mechanisms naturally compatible for feature-level knowledge transfer. In contrast, YOLO11 is a CNN-based detector with a substantially different feature hierarchy, inductive bias, and detection head. As a result, directly matching intermediate features from a DETR-style teacher to a YOLO-style student is less effective.
> >
> > This observation is also consistent with previous studies [1, 2] on cross-architecture knowledge distillation, which report that feature distillation becomes considerably more challenging when the teacher and student adopt different architectures.
> >
> > **References**
> >
> > [1] Liu, Y., et al. *Cross-Architecture Knowledge Distillation.* Proceedings of the Asian Conference on Computer Vision (ACCV), 2022.
> >
> > [2] Zheng, X., et al. *Distilling Efficient Vision Transformers from CNNs for Semantic Segmentation.* Pattern Recognition, 158 (2025): 111029.

---

### Review · Reviewer_R1kX · 2026-04-26

**Summary Of Contributions:**

The paper tackles how to adapt the ViT-based framework for edge-oriented dense prediction tasks, such as object detection, segmentation, and pose estimation. The paper argues the key to improving the compact version the ViT's performance is through distillation. The distillation should also be task-specific. Specifically, this paper uses a DINOv3 pre-trained on object detection and then distills detection-oriented DINOv3's representation into small ECViT student backbones. After the distillation, the compact ECViT backbone can be used for different dense prediction tasks, eg, segmentation.

The strength:
1. Strong empirical performance across three COCO dense prediction tasks under relatively compact parameters and FLOPs.
2. The reported results suggest that compact ViTs can be competitive with CNN or YOLO-style models when paired with task-specialized distillation and appropriate architectural modifications.
3. The paper includes useful ablations on the teacher scale, teacher task adaptation and other design choices like feature-alignment depth, optimizer etc.

The weakness:
1. The major weakness is that the conceptual novelty is limited. Most individual components and their effectiveness are quite known by the community such as distillation from large-pretrained model, feature alignment, multi-scale feature construction from ViT and DETR style decoders. The contribution is primarily in the integration and show the empirical validation of this combination of techniques works for ViT. There seems limited new knowledge that is provided to the community.
2. Some claims require more careful qualification since the latency is not consistently better than the competing methods and the method relies on a DINOv3 teacher pre-trained on large-scale external data, which is not surprising to have better results.

**Audience:**

Yes

**Audience Explanation:**

Yes, the paper would be of interest to researchers and practitioners working on efficient dense prediction, compact ViTs, knowledge distillation, and deployment-oriented vision models.

**Broader Impact Concerns:**

I do not see major broader-impact concerns.

**Claims And Evidence:**

Yes

**Claims Explanation:**

The paper provides some convincing evidence for its main empirical claim: compact ViTs can perform competitively on the dense prediction tasks when using the task-specific DINOv3's feature distillation. The COCO results for detection, pose estimation, and instance segmentation are strong, and the ablations on teacher scale, teacher adaptation, and register tokens support several design choices.

However, some claims should be better qualified. The novelty of task-specialized distillation appears incremental, since much of the gain comes from using DINOv3-style features rather than from the task adaptation alone. The "edge" claim is also only partially supported, since the models are compact in parameters and FLOPs but are not consistently faster than YOLO-style baselines. The paper should also more clearly distinguish "only COCO annotations" from the use of a large externally pretrained DINOv3 teacher.

**Requested Changes:**

1. The control observation that task-specific distillation from a stronger teacher improves a compact student is well established. Therefore, the paper's novelty should not be framed around this observation alone. The authors should clarify that their contribution lies in the specific integration of the detection-adapted DINOv3 teacher, compact ViT student design, and cross-task prediction evaluation.

2. Separate the DINOv3 effect.  The paper’s central claim is about task-specialized distillation, but a substantial portion of the gain appears to come from using DINOv3-style distillation. To isolate the value of proposed DINOv3-based task-specialized distillation, the authros should also compare against distillation from a strong non-DINO detector teacher, such as high-performaning Co-DETR, using the same student architecture and training budget.

3.    The paper claims that the detection-distilled representation transfers to instance segmentation and pose estimation. This would be more convincing if the authors included initialization ablations for ECInsSeg and ECPose, analogous to the ECDet ablation

---

> ### Author Response · Authors · 2026-07-07
>
> We sincerely thank Reviewer R1kX for the thoughtful and constructive feedback. We are pleased that the reviewer recognizes the strong empirical performance of ECDet across multiple dense prediction tasks, as well as the comprehensive ablation studies supporting our design choices. Below, we address each concern in detail.
>
>
> > The conceptual novelty is limited since most individual components are well known.
>
> We agree that the individual components in our framework—knowledge distillation, feature alignment, multi-scale feature construction, and DETR-style decoders—are not novel in isolation. We have discussed these related directions in the paper and will further clarify this positioning in the revised manuscript.
>
> Our contribution is not to claim novelty for each individual component. Instead, **our main contribution is the complete framework showing that a compact ViT can become practical for edge dense prediction.**
>
> This is important because current real-time edge systems are still largely dominated by CNN-based detectors such as YOLO, while compact ViTs generally require much larger backbones or substantially heavier pretraining to achieve competitive performance. As discussed in the Introduction, this has limited the practical deployment of ViTs on edge devices.
>
> Our results demonstrate that this gap can be significantly reduced by combining:
>
> - a task-specialized teacher,
> - an efficient compact ViT backbone, and
> - an edge-oriented dense prediction framework.
>
> Importantly, the learned backbone is not limited to object detection. The same distilled representation transfers effectively to instance segmentation and human pose estimation, providing strong accuracy-efficiency trade-offs across all three dense prediction tasks. This suggests that the learned compact ViT representation can serve as a shared backbone for multiple edge vision applications.
>
> Overall, we believe this work provides a practical and empirically validated recipe for deploying compact ViTs in edge dense prediction, while challenging the common assumption that edge vision should primarily rely on CNN-based backbones.
>
>
> > The "edge" claim should be better qualified because latency is not consistently better than competing methods.
>
> We agree that ECDet is not always the fastest detector in terms of absolute latency, and we will revise the wording to make the "edge" claim more precise.
>
> By **edge**, we do not mean that ECDet is always the fastest model. Rather, we refer to the overall deployment efficiency under edge constraints, where multiple factors are important:
>
> - **Detection accuracy**
> - **Parameter count**
> - **FLOPs**
> - **Memory and storage cost**
> - **Real-time inference latency**
>
> As already discussed in the manuscript, current hardware and software stacks are considerably more optimized for CNN-based models than for ViT-based models [1], which partially explains the remaining latency gap.
>
> This distinction is particularly important for real edge deployment. For example, commercial NPUs such as the Rockchip RK3568 provide only about 1 TOPS INT8 computing capability ((https://www.rock-chips.com/uploads/pdf/2022.8.26/192/RK3568%20Brief%20Datasheet.pdf). On such devices, model size and computational complexity directly affect deployment feasibility through memory usage, storage requirements, compiler support, and hardware-specific optimizations. Furthermore, the Rockchip RKNN Model Zoo (https://github.com/airockchip/rknn_model_zoo#model-performance-benchmarkfps) reports noticeably different inference speeds for the same models across different hardware platforms, illustrating that practical latency depends heavily on deployment backends rather than model architecture alone.
>
> Our main claim is therefore about the overall accuracy-efficiency trade-off. ECDet-S achieves 51.7 AP with 10M parameters, 26 GFLOPs, and 5.41 ms latency on T4, which is about 185 FPS. **Some YOLO-style models are faster, but ECDet-S gives higher accuracy under a similar parameter and FLOP budget**. At a larger scale, **ECDet-L achieves 57.0 AP with 10.49 ms latency, while YOLO26-X achieves 56.9 AP with 11.65 ms latency**. In this high-accuracy range, ECDet-L is both slightly more accurate and faster.
>
> These results show that ECDet consistently provides competitive or superior accuracy under similar computational budgets, while remaining practical for real-time deployment.
>
> ### References
> [1] Xia, X., Li, J., Wu, J., et al. TRT-ViT: TensorRT-oriented Vision Transformer. arXiv preprint arXiv:2205.09579, 2022.

---

> > ### Author Response · Authors · 2026-07-07
> >
> > > The paper should more clearly distinguish "only COCO annotations" from the use of an externally pretrained DINOv3 teacher.
> >
> > We thank the reviewer for pointing this out. We agree that our wording should be clearer. When we say “only COCO annotations,” we mean that the downstream detector is trained with COCO detection annotations only. However, the distillation stage does use additional unlabeled images: ImageNet-1K train images and COCO train2017 images. No ImageNet labels are used during distillation. We will make the data usage clearer in the revised manuscript:
> >
> > - **Detector training:** uses COCO train2017 images with COCO bounding-box annotations.
> > - **Distillation:** uses ImageNet-1K train images and COCO train2017 images without labels.
> > - **DINOv3:** is an externally pretrained model used to initialize the backbone of our teacher. It is not directly used as the final teacher detector. We first adapt it to COCO detection to obtain ECDet-Teacher, and then use this teacher to distill the compact student backbone.
> >
> > Therefore, a more precise statement is that our downstream dense prediction models are trained with COCO task annotations, while the distillation stage uses additional unlabeled images and an externally pretrained DINOv3 initialization. We will revise the paper to clearly separate these three parts: external pretrained initialization, unlabeled distillation images, and human annotations.
> >
> > > Separate the DINOv3 effect. The paper’s central claim is about task-specialized distillation, but a substantial portion of the gain appears to come from using DINOv3-style distillation
> >
> > To better separate the effect of DINOv3 from the effect of task-specialized distillation, we added a comparison with a state-of-the-art non-DINO detector teacher, Co-DETR-Swin-S [2], using the same student architecture ECDet-S and the same training budget including distillation and detection.
> >
> > |   |   |   |
> > |---|---|---|
> > |Model|Teacher / Initialization|AP|
> > |ECDet-S|Random initialization|46.6|
> > |ECDet-S|Co-DETR-Swin-S teacher|50.4|
> > |ECDet-S|DINOv3-S teacher|50.7|
> > |ECDet-S|ECDet-Teacher-S|51.7|
> >
> > The results reveal three observations:
> >
> > - **Distillation from a strong non-DINO detector teacher is helpful.** Distillation from Co-DETR-Swin-S improves AP from **46.6** to **50.4**.
> > - **DINOv3 provides stronger generic visual representations.** Using DINOv3-S further improves AP to **50.7**, outperforming Co-DETR-Swin-S under the same training budget.
> > - **Task adaptation contributes additional gains.** Our detection-adapted teacher achieves **51.7 AP**, improving performance by **+1.0 AP** over DINOv3-S and **+1.3 AP** over Co-DETR-Swin-S.
> >
> > Therefore, the performance gain cannot be explained solely by DINOv3 pretraining. While DINOv3 provides a strong initialization, adapting the teacher to the downstream detection task consistently brings additional improvements.
> >
> > > This would be more convincing if the authors included initialization ablations for ECInsSeg and ECPose, analogous to the ECDet ablation
> >
> > We appreciate this valuable suggestion. Following the reviewer's recommendation, we added initialization ablations for both **ECInsSeg** and **ECPose**.
> >
> > The instance segmentation results are shown below.
> >
> > | Model | Initialization | AP |
> > |:------:|:--------------|:--:|
> > | ECInsSeg-S | Random | **38.0** |
> > | ECInsSeg-S | Distilled backbone | **42.8** |
> > | ECInsSeg-S | Detection backbone | **43.0** |
> > | ECInsSeg-L | Distilled backbone | **46.4** |
> > | ECInsSeg-L | Detection backbone | **47.1** |
> >
> > The human pose estimation results are shown below.
> >
> > | Model | Initialization | AP |
> > |:------:|:--------------|:--:|
> > | ECPose-S | Random | **58.6** |
> > | ECPose-S | Distilled backbone | **68.4** |
> > | ECPose-S | Detection backbone | **68.9** |
> > | ECPose-L | Distilled backbone | **72.5** |
> > | ECPose-L | Detection backbone | **73.5** |
> >
> > These results show a consistent trend across both tasks.
> >
> > - The distilled backbone provides a strong initialization.
> > - ECInsSeg-S improves from **38.0** to **42.8 AP** (**+4.8 AP**).
> > - ECPose-S improves from **58.6** to **68.4 AP** (**+9.8 AP**).
> >
> > - The detection-trained backbone further improves transfer performance.
> > - ECInsSeg-S improves from **42.8** to **43.0 AP**.
> > - ECPose-S improves from **68.4** to **68.9 AP**.
> >
> > ### References
> >
> > [1] Xia, X., Li, J., Wu, J., *et al.* **TRT-ViT: TensorRT-oriented Vision Transformer.** *arXiv preprint arXiv:2205.09579*, 2022.
> >
> > [2] Zong, Z., Song, G., and Liu, Y. **DETRs with Collaborative Hybrid Assignments Training.** *Proceedings of the IEEE/CVF International Conference on Computer Vision (ICCV)*, 2023.

---

### Review · Reviewer_Y2uq · 2026-06-24

**Summary Of Contributions:**

In this paper, the authors study the problem of visual detection (object detection, instance segmentation and pose estimation) in low-resource settings. To be specific, they propose EdgeCrafter as a multi-stage framework which distills and strong but lightweight backbone from a strong teacher and allows using the same backbone for various visual detection tasks. The experiments show significant improvements of EdgeCrafter with different scales on different tasks.

Strengths:
+ Lightweight visual detection is an important problem in computer vision.
+ The proposed approach is very sensible and sound.
+ A better Pareto frontier for AP vs. model size & inference speed is provided compared to the state of the art.

**Audience:**

Yes

**Audience Explanation:**

Visual detection is a hot topic in ML/AI literature.

**Broader Impact Concerns:**

Not present and not necessary.

**Claims And Evidence:**

No

**Claims Explanation:**

Weaknesses:

1. A major concern is whether the improvements are due to the well-designed architecture, or they are simply a byproduct of a strong teacher. I would suggest the following:

1.1. What happens if you train the EdgeCrafter student architecture using standard supervised pre-training, without the distillation pipeline? This analysis would be helpful to see if your architectural modifications are inherently robust, or they are entirely dependent on DINOv3's representations.

1.2. If we apply your exact Task-Specialized Distillation pipeline to an existing, well-optimized edge backbone (e.g., MobileNetV4, EfficientViT, or a small YOLO backbone), does it achieve similar or better Pareto-efficiency than the EdgeCrafter student? If it does, your paper’s core contribution is a distillation framework, not an architectural framework.

1.3. It would be helpful also to compare the proposed simple interpolation and linear projection against a traditional FPN/PANet.

2. The paper compares its T4 latency numbers against real-time detectors like RT-DETR and the YOLO family. In the interest of fair comparison, were the competing CNN architectures benchmarked using the exact same environment parameters (i.e., torch.compile, TensorRT v10.6, and FP16 precision)?

3. Using a foundation model as a teacher provides helpful supervisory signals, but it introduces massive training-time overhead. The overhead can be a limiting factor for the practical adoptability of the framework for custom downstream datasets.

**Requested Changes:**

Please see Weaknesses above.

---

> ### Author Response · Authors · 2026-07-07
>
> We appreciate the reviewer for the constructive comments. The reviewer raises several important questions regarding the source of the performance gains, the generality of the proposed distillation framework, and the fairness of the experimental comparisons. These suggestions help us better clarify the scope of our contributions. We respond to each point below.
>
> ---
>
> > What happens if you train the EdgeCrafter student architecture using standard supervised pre-training, without the distillation pipeline
>
> We thank the reviewer for the suggestion. We agree that it is important to separate the effect of the architecture from the effect of the teacher.
>
> We first trained the EdgeCrafter student architecture without our distillation pipeline, using the standard COCO detection training protocol. This model achieves 46.6 AP. We also initialized the same ECViT-Tiny backbone with standard supervised ImageNet-1K pretraining, which gives 46.7 AP, almost the same as random initialization. In addition, we tried the Google-released ViT-Tiny supervised pretrained on ImageNet-21K (https://huggingface.co/timm/vit_tiny_patch16_224.augreg_in21k), which only gives 42.2 AP in our detection setting.
>
> | Model | Pretraining / Initialization | AP |
> |:------:|:----------------------------|:--:|
> | ViT-Tiny | ImageNet-21K supervised pretraining | 42.2 |
> | ECDet-S | Random initialization | 46.6 |
> | ECDet-S | ImageNet-1K supervised pretraining | 46.7 |
> | ECDet-S | Task-specialized distillation | 51.7 |
>
> These results show two things.
>
> First, the EdgeCrafter architecture itself is important. Even without distillation, ECDet-S reaches 46.6 AP, clearly better than the original ViT-Tiny baseline with ImageNet-21K supervised pretraining.
>
> Second, standard supervised pretraining is not enough for this compact ViT setting. ImageNet-1K supervised pretraining only improves ECDet-S from **46.6** to **46.7 AP**. In contrast, our task-specialized distillation improves the same architecture to **51.7 AP**, giving a **+5.1 AP** gain.
>
> Therefore, the improvement does not come only from the architecture or only from the strong teacher. The architecture gives a strong compact student, and task-specialized distillation provides the additional representation needed to make it competitive.
>
> ---
> >  If we apply your exact Task-Specialized Distillation pipeline to an existing, well-optimized edge backbone, does it achieve similar or better Pareto-efficiency than the EdgeCrafter student.
>
> We thank the reviewer for the insightful suggestion. To test whether the gain mainly comes from the distillation pipeline or from the EdgeCrafter student design, we applied the same task-specialized distillation pipeline to an existing edge-oriented backbone, EfficientViT-M3 [1].
>
> | Backbone | Pretraining / Initialization | AP |
> |:---------:|:----------------------------|:--:|
> | EfficientViT-M3 | ImageNet-1K supervised pretraining | 42.7 |
> | EfficientViT-M3 | Task-specialized distillation | 44.7 |
> | ECDet-S | Task-specialized distillation | 51.7 |
>
> The distillation pipeline improves EfficientViT-M3 from **42.7 AP** to **44.7 AP**, giving a **+2.0 AP** gain. This shows that our distillation strategy can also help other compact backbones. However, the final performance is still much lower than ECDet-S, which reaches **51.7 AP** under the same distillation framework. This suggests that distillation alone cannot explain the full improvement.
>
> We believe the student architecture is important. EdgeCrafter and the teacher are both ViT-style models, so their feature representations are more naturally aligned during distillation. EfficientViT has a different structure, so direct feature-level distillation may not transfer the teacher representation as effectively. Therefore, our conclusion is that the gain comes from both parts: the task-specialized distillation pipeline and the EdgeCrafter student architecture. Distillation helps, but the architecture is important for fully using the transferred representation.
>
> ---
>
> >  It would be helpful also to compare the proposed simple interpolation and linear projection against a traditional FPN/PANet.
>
> Our multi-scale feature construction is already close in spirit to a lightweight PAN-style design. Starting from the compact ViT feature map, we generate multi-scale features through interpolation and linear projection, and then pass them to the detection encoder/decoder. The goal is similar to FPN/PANet: to provide multi-scale features for dense prediction. The main difference is that we use a simpler and lighter implementation, because the backbone is a compact ViT and we want to avoid adding a heavy CNN-style feature pyramid.
>
>
> **References**
>
> [1] Liu, Xinyu, et al. *EfficientViT: Memory Efficient Vision Transformer with Cascaded Group Attention.* Proceedings of the IEEE/CVF Conference on Computer Vision and Pattern Recognition (CVPR), 2023.

---

> > ### Author Response · Authors · 2026-07-07
> >
> > > In the interest of fair comparison, were the competing CNN architectures benchmarked using the exact same environment parameters (i.e., torch.compile, TensorRT v10.6, and FP16 precision)?
> >
> > Yes. All models, including CNN-based and transformer-based detectors, are benchmarked under the same inference environment. Specifically, we measure latency on the same NVIDIA T4 GPU, with batch size 1, FP16 precision, and TensorRT v10.6. For YOLO-style models, the reported latency includes NMS/post-processing, following common real-time detection practice. We use the same benchmarking protocol for all compared methods to make the latency comparison as fair and consistent as possible.
> >
> > > Using a foundation model as a teacher provides helpful supervisory signals, but it introduces massive training-time overhead. The overhead can be a limiting factor for the practical adoptability of the framework for custom downstream datasets.
> >
> > We thank the reviewer for raising this concern. For ECDet-M, the full training pipeline has two stages.
> >
> > First, we distill the compact backbone using unlabeled ImageNet-1K and COCO images. This stage uses 2 RTX 4090 GPUs for about **17 hours**, corresponding to **34 GPU hours**.
> >
> > Second, we train the detector on COCO using **4 RTX 4090 GPUs** for about **24 hours**, corresponding to **96 GPU hours**.
> >
> > In total, **ECDet-M** requires approximately **130 GPU hours**.
> >
> > | Method | Pre-training / Distillation Data | COCO Detection Epochs | GPU Hours | AP |
> > |:------:|:--------------------------------:|:---------------------:|:---------:|:--:|
> > | YOLO11-M | - | 600 | ~160 | 51.2 |
> > | RT-DETRv4-M | IN-22K + COCO | 102 | ~190 | 53.5 |
> > | ECDet-M | IN-1K + COCO | 74 | 130 (96 + 34) | 54.3 |
> >
> > Although our method introduces an additional distillation stage, the total training cost remains comparable to common real-time detector training pipelines. For example, **ECDet-M** achieves **54.3 AP** with approximately **130 GPU hours**, while **YOLO11-M** uses about **160 GPU hours** to achieve **51.2 AP**, and **RT-DETRv4-M** uses about **190 GPU hours** to achieve **53.5 AP**. **Note that all GPU-hour statistics are measured using the same 4 × RTX 4090 GPUs.**
> >
> > Therefore, when adapting to custom downstream datasets, the overall training cost of our method is comparable to, and in some cases even lower than, that of widely used real-time detectors.